# An energizing role for motivation in information-seeking during the early phase of the COVID-19 pandemic

Yaniv Abir [1✉], Caroline B. Marvin[1], Camilla van Geen [2,3], Maya Leshkowitz[4], Ran R. Hassin[5,7] & Daphna Shohamy[1,2,6,7]

The COVID-19 pandemic has highlighted the importance of understanding and managing information seeking behavior. Information-seeking in humans is often viewed as irrational rather than utility maximizing. Here, we hypothesized that this apparent disconnect between utility and information-seeking is due to a latent third variable, motivation. We quantified information-seeking, learning, and COVID-19-related concern (which we used as a proxy for motivation regarding COVID-19 and the changes in circumstance it caused) in a US-based sample ($n = 5376$) during spring 2020. We found that self-reported levels of COVID-19 concern were associated with directed seeking of COVID-19-related content and better memory for such information. Interestingly, this specific motivational state was also associated with a general enhancement of information-seeking for content unrelated to COVID-19. These effects were associated with commensurate changes to utility expectations and were dissociable from the influence of non-specific anxiety. Thus, motivation both directs and energizes epistemic behavior, linking together utility and curiosity.

[1] Department of Psychology, Columbia University, New York, NY, USA. [2] Zuckerman Mind Brain Behavior Institute, Columbia University, New York, NY, USA. [3] Department of Psychology, University of Pennsylvania, Philadelphia, PA, USA. [4] Department of Cognitive Sciences, The Hebrew University of Jerusalem, Jerusalem, Israel. [5] Department of Psychology and The Federmann Center for the Study of Rationality, The Hebrew University of Jerusalem, Jerusalem, Israel. [6] Kavli Institute for Brain Science, Columbia University, New York, NY, USA. [7]These authors contributed equally: Ran R. Hassin, Daphna Shohamy. ✉email: yaniv.abir@columbia.edu

The disputed ability of humans to rationally sift through the trove of information they encounter daily has inspired a conflicted debate[1–3]. A common view is that humans are ill-equipped to scale an informational landscape formed with the goal of capturing and exploiting attention[4–7]. Thus, useless or even nefarious information receives a wide audience, as it plays to the inherent irrationality of human information-seeking[8–12].

This prevalent conviction, that human information-seeking is mired by irrationality, is grounded in a long philosophical tradition (including Cicero, Philo, Edmund Burke, and Heidegger[13,14]) and is reflected in a wide array of cognitive research positing that maximization of instrumental value is not a sufficient explanation for human information-seeking[14–16]. Proponents of this view grant that humans can select the most useful questions when constrained by clear and immediate goals—such as when preparing for a test[17–19] (cf[20].). But they claim that usefulness does not guide unconstrained information-seeking, which is instead shaped by curiosity—an intrinsic drive for knowledge for its own sake, ostensibly without regard of utility, or even in spite of informational disutility[14–16,21–25]. Curiosity as a goal-independent drive is claimed to be necessary to explain why humans commonly seek more information than is prescribed by economic utility[14,26].

However compelling, the designation of curiosity as a goal-independent drive faces considerable challenges. Curiosity, as examined by specifically crafted non-instrumental tasks, has been found to predict better learning[27,28]. This better learning, seen also in educational, developmental and robotics research[29–31], matches William James' description of curiosity as "the impulse towards better cognition"[32], a notion that cannot parsimoniously be divorced from goal attainment[33,34]. Furthermore, a striking similarity was found between the neural correlates of processing non-instrumental information, and that of processing instrumental information, or even reward[22,27,35,36].

We hypothesized that instead of positing a separate drive, the apparent irrationality in information-seeking could be explained by the operation of a crucial third variable: motivation. Researchers have heretofore focused on the incentive value of information, predicting that people should generally seek to maximize their informational intake[14,16,37]. However, we reasoned that just as a food reward does not have the same utility for a hungry subject and a satiated one[38–40], the same informational content should elicit varying expectations of utility from participants in different motivational states. This prediction is grounded in learning theories of motivation, defining a motivational state as the mapping between potential actions and their value[38,39]. By measuring a person's motivational state, as influenced by their personal circumstance, it should be possible to infer their information-seeking goals and to test whether their behavior is cost-benefit rational given their goals.

The onset of the coronavirus pandemic (COVID-19) in 2020 presented the opportunity to examine naturalistic and meaningful information-seeking and its link to motivation and utility. Suddenly, many people were curious about the virus and its epidemiology, which was now motivationally relevant as well as intellectually interesting. Additionally, differences in age, geography and circumstance resulted in considerable, and measurable, individual differences in the personal relevance of information regarding COVID-19. Together, these features allowed us to study ecologically valid information-seeking, while measuring naturally occurring variations in utility and motivation.

Furthermore, COVID-19, being the first global pandemic of the information age, demonstrated a pressing need for a theory of information-seeking that can predict human behavior. Throughout this global crisis, policymakers face challenges in managing the seeking and sharing of information by the general public. Indeed, the WHO chose the term "infodemic" as a banner phrase championing information-seeking as a public health policy goal[41,42]. As cognitive scientists, however, we find ourselves limited in providing applied insights on this matter, lacking a theoretical framework for information-seeking under such conditions.

To investigate the role of motivation and utility in information-seeking we collected data from 5376 participants across the United States, using Amazon Mechanical Turk to sample the population twice a week between March 11th and May 7th, 2020. We measured people's epistemic behavior: their choices to seek answers to questions (both related and unrelated to COVID-19), their subsequent satisfaction with revealed answers, and their memory recall for that information one week later (see Methods). In addition, we collected judgments of question usefulness, a proxy for utility expectations. Importantly, we also assessed participants' specific motivational states regarding COVID-19 using a questionnaire-based measure of COVID-19 concern. To corroborate that this measure is specifically related to COVID-19, we also measured participants' non-specific anxiety as a control for general affective differences. Our use of these two measures is predicated on the view that affective states are motivational states[43,44]. Altogether, this is the first study to examine epistemic behavior in tandem with utility expectations and motivational states, with the goal of uncovering the associations between these constructs.

Our main hypothesis concerned the relationship between motivation, utility expectations, and epistemic behavior. A rational analysis of information-seeking prescribes a *directing* role for motivation, whereby specific motivational states direct participants to seek information in specific content domains[33,37,45,46]. Motivation, under this account, is defined as the mapping between information and utility, and so should exert its directing effect by changing utility expectations[38–40]. In our case, people with higher COVID-19 concerns should tend to seek more COVID-19-related information relative to general information and judge such information to be more useful. Based on these theories, we further predicted that such a directing effect of COVID-19 concern should be independent of the effect of non-specific anxiety our participants may be experiencing.

Established theories of reward-based behavior, if allowed to extend to epistemic behavior, make a second prediction regarding the role of motivation in information-seeking. Such theories, formulated over half a century ago, suggest that shifts in motivation not only direct behavior but may also *energize* it by changing the average utility of action[38,39,47]. Hungry rats tend to work more vigorously not only for food but also for water because by virtue of raising the utility of gaining food, hunger raises the average utility of instrumental action. Correspondingly, people who are acutely motivated to seek answers to questions in one domain, for example, if they are highly concerned about COVID-19, should be more likely to seek information of any kind, even answers to questions that bear no relevance to COVID-19. They should also report elevated utility expectations for gaining answers to questions. This prediction is not normatively prescribed by rational analysis but is rather the result of the approximate way such computations are implemented by cognitive systems (see Discussion).

Lastly, theories of curiosity as a non-instrumental intrinsic drive can accommodate the predictions above, but only for information that is commonly considered useful. Since humans so often seek information that is conventionally considered useless, such as answers to trivia questions, it is claimed that utility may not be considered a general predictor of information-seeking[14,36,48]. In contrast, mechanistic theories of curiosity[33,46] and motivation[34,39,49] posit that all behavior is essentially goal-based and so driven by expectations of utility, even if the prospect of an answer being rewarding is very remote or abstract[34,50]. As

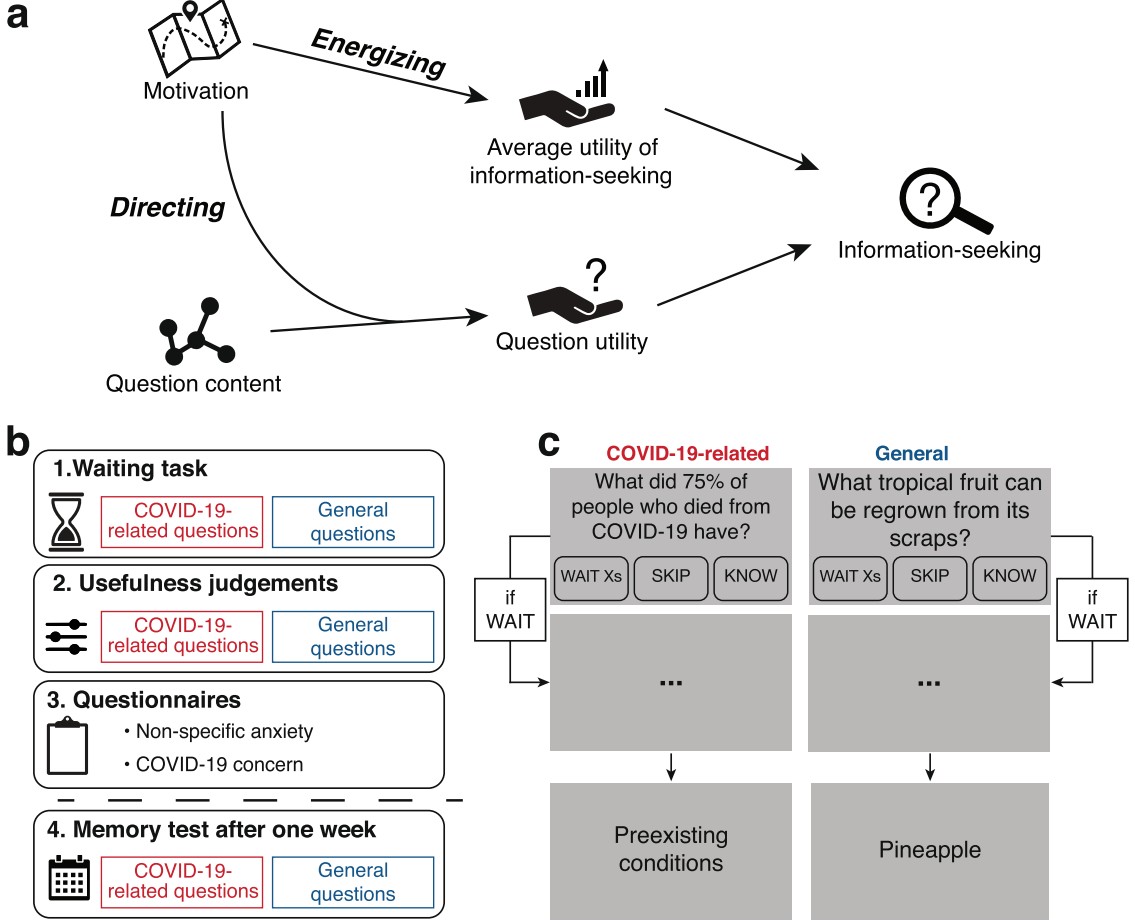

**Fig. 1 Measuring the two roles of motivation in information-seeking. a** Based on rational theories of motivation, we postulate a dual role for motivation in modulating information-seeking. First, specific motivational states direct information-seeking by assigning greater utility to questions with relevant content. Motivational states also exert an energizing influence on information-seeking, by changing the average utility of seeking information. **b** To measure these effects, we employed the waiting task, in which participants decided whether to seek answers to questions that were either related to COVID-19 or not. They then rated a separate set of questions on their usefulness, a proxy for utility expectations. Their specific motivational state regarding COVID-19 was assessed with the COVID-19 concern questionnaire, which was preceded by a non-specific anxiety questionnaire as a control measure. One week later, participants were asked to recall the answers they had chosen to pursue. **c** On each trial of the waiting task, participants were presented with a question. If they already knew the answer, they could indicate it as such. Otherwise, if they decided to wait a specified duration, (ranging 4–16 s) they were presented with the answer. They were then asked to rate their satisfaction with the answer and, one week later, to report their memory (not pictured).

such, they make no exception for trivia, and predict that the pursuit of answers to trivia questions should depend on utility and motivation just like for any other class of questions. To adjudicate between these contrasting claims regarding the usefulness of trivia, our stimulus set included trivia questions as well as questions conventionally viewed as useful.

### Results

Epistemic behavior was assessed in 5376 participants using the waiting task (Fig. 1). On each trial, participants were presented with a question that was either related or unrelated to COVID-19. Participants chose whether to wait a variable number of seconds for the answer to the question. Their choices to wait rather than skip, given the different wait durations, serve as our information-seeking measure. Participants further rated their satisfaction with the answers read and had to recall them in a test one week later. These three measures of epistemic behavior—information-seeking, satisfaction, and recall, were thoroughly validated in the previous studies[27,28,36]. We sought to explain patterns of epistemic behavior based on participants' levels of COVID-19 concern, as detailed above.

**The directing effect of motivation.** We first evaluated the hypothesis that information-seeking is directed by participants' motivational state and, as such, is goal-rational. Thus, we focused on the interaction effect of COVID-19 concern and question type on waiting for answers. We found that people reporting higher COVID-19 concern were more likely to seek answers to COVID-19-related questions relative to general questions (interaction $b = 0.11$, 95% PI = [0.08, 0.14]; Fig. 2a, Eqn. S6). This is consistent with the notion that motivation has a directing effect: a change in domain-specific motivation should impact information-seeking in that domain.

We further hypothesized that COVID-19 concern should operate on information-seeking by increasing the expected utility of questions related to COVID-19. Indeed, we find that higher COVID-19 concern is associated with higher usefulness judgments of COVID-19-related questions relative to general questions (interaction $b = 0.10$, 95% PI = [0.07, 0.12]; Fig. 2b, Eqn. S10). Furthermore, a mediation analysis revealed a significant indirect effect of the interaction of COVID-19 concern and question type (COVID-19-related/general) on waiting choices, via usefulness judgments $b = 0.02$, 95% PI = [0.002, 0.04], with 22.27% of the effect mediated, 95% PI = [1.71%, 49.86%] (Fig. 2c, Eqn. S11–12).

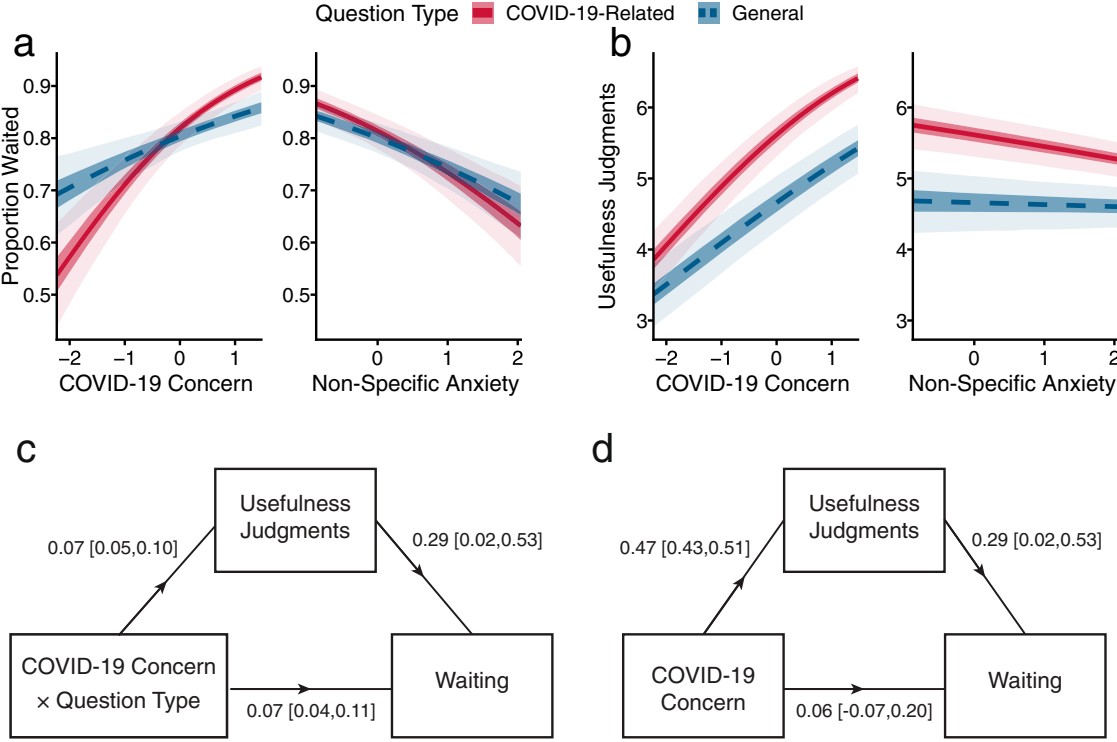

**Fig. 2 Motivation is associated with a directed and energized pattern of information-seeking. a** Higher COVID-19 concern is associated with more waiting for COVID-19-related questions, relative to general questions. It is also associated with more waiting for general questions (left panel). Non-specific anxiety, on the other hand, negatively predicts waiting for all questions (right). **b** Participants reporting higher COVID-19 concern judge questions as more useful, especially COVID-19-related questions (left panel). Participants reporting higher levels of non-specific anxiety tend to judge questions as less useful, especially COVID-19-related questions (right). **a**, **b** Lines denote mean posterior prediction; dark shaded areas mark 50% PIs, and light areas 95% PIs. **c** Data are consistent with our hypothesis that the directing effect of COVID-19 concern on information-seeking should be mediated via expectations of utility. We find a significant indirect effect of the interaction between COVID-19 concern and question type on waiting, mediated by judged usefulness. COVID-19 concern is associated with the relative usefulness of COVID-19-related vs. general information and thus associated with waiting choices for COVID-19-related vs. general information. **d** We also find a significant indirect main effect of COVID-19 concern, via judged usefulness, on waiting. This is consistent with motivational theories predicting an energizing effect of COVID-19 concern on information-seeking, mediated via expectations of utility. 95% PIs are given in brackets. Source data are provided as Source Data files.

**The energizing effect of motivation**. We continued to assess whether the data support the predicted energizing effect of motivation on information-seeking. In our experimental design, an energizing effect should be manifested as the main effect of COVID-19 concern on information-seeking for all questions, mediated by each participant's average usefulness judgments. Indeed, higher COVID-19 concern is associated with increased waiting not only for COVID-19-related questions, but also for general questions (simple effect $b = 0.17$, 95% PI = [0.09, 0.26]; Fig. 2a, Eqn. S6), as well as with a general rise in usefulness judgments (main effect $b = 0.51$, 95% PI = [0.47,0.55]; Fig. 2b, Eqn. S10). The mediation analysis confirmed a significant indirect effect of COVID-19 concern on waiting via usefulness judgments $b = 0.14$, 95% PI = [0.01, 0.25], (69.27% of the effect mediated 95% PI = [5.38%, 151.16%]; Fig. 2d, Eqn. S11–12).

**Answer satisfaction reflects motivational changes to utility and determines subsequent information-seeking**. The link between utility and information-seeking should be observable not only when participants are expecting an answer, but also after they receive it. Thus, participants reported satisfaction with each answer poses a second test for the predicted relationship between motivation and epistemic behavior. Indeed, we found that COVID-19 concern was associated with the directing and energizing of participants' reported satisfaction with the answers. Mirroring the reported effects on information-seeking, we

observe that people high in COVID-19 concern were more satisfied with all answers (main effect $b = 0.43$, 95% PI = [0.38,0.48]), but especially with COVID-19-related answers (interaction $b = 0.08$, 95% PI = [0.06,0.11]; Fig. 3a, Eqn. S7).

Finally, we tested whether the subjective experience of utility on one trial might shape the decision to wait for information on the next trial. The purpose of this analysis was to determine whether a surprisingly satisfying (or dissatisfying) answer might play a causal role in later information-seeking. To test this possibility, we quantified how utility expectations for a question, measured by usefulness ratings, and participants' actual satisfaction with an answer jointly influence information-seeking in the next trial. In line with predictions from reinforcement learning theory[28,51–54], we found that high answer satisfaction predicted more waiting on the subsequent trial $b = 0.10$, 95% PI = [0.05, 0.15], while high usefulness expectations predicted less waiting on the subsequent trial $b = -0.09$, 95% PI = [-0.13,-0.05] (Eqn. S13). Together these terms form the difference between satisfaction and expectation, or the prediction error[51,52], which determines future information-seeking (Fig. 4a, b). Crucially, such an influence of prediction errors suggests that participants are using an average value of information-seeking, carried across trials, to guide their information-seeking[53,55], a central tenet of our framework (Fig. 1a).

**Control analyses**. To make claims about utility and motivation in the context of COVID-19, it is important to distinguish between

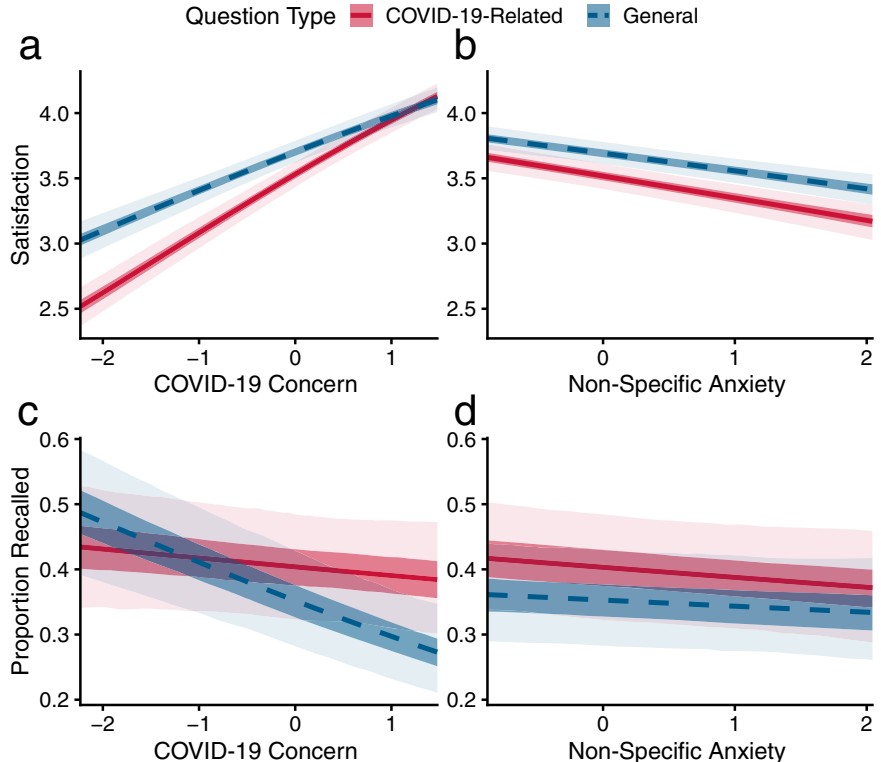

**Fig. 3 Motivation correlated with subsequent satisfaction with answers to questions and subsequent memory for the answers. a** Greater COVID-19 concern is associated with more self-reported satisfaction, especially for COVID-19-related questions. **b** In contrast, non-specific anxiety is associated with a reduction in satisfaction for both question types. **c** COVID-19 concern is associated with poorer memory for general information, while information related to COVID-19 is spared. **d** Non-specific anxiety is not significantly associated with memory. Lines denote mean posterior prediction from multilevel regression models, dark shaded areas mark 50% PIs, and light areas 95% PIs. Source data are provided as Source Data files.

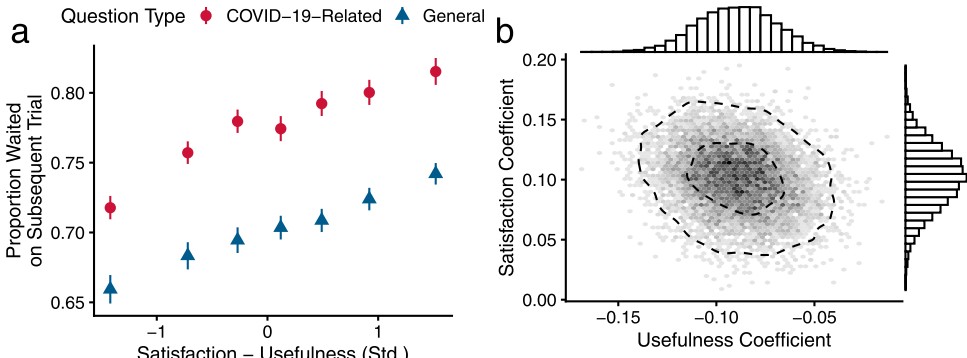

**Fig. 4 Prediction errors influence information-seeking on the subsequent trial. a** Prediction errors, defined as the difference between actual satisfaction with answers minus usefulness expectations engendered by the question, determined waiting for choices on the subsequent trial. The influence of prediction errors testifies to participants' use of the average value of information in making waiting choices. Data presented as mean values ±1 SEM; $n = 4754$ participants. **b** Posterior distribution of the coefficients for usefulness expectations and satisfaction in a multilevel regression model predicting waiting choices on the subsequent trial. The satisfaction coefficient is significantly positive, while the usefulness coefficient is significantly negative. Together, the two coefficients reflect the influence of prediction errors. Dashed contours denote 95 and 50% posterior densities. Source data are provided as Source Data files.

specific shifts in motivational states and between a more general, non-specific form of anxiety. Accordingly, we obtained separate measures of these two constructs and found that the effects of COVID-19 concern on information-seeking do not appear to be due to non-specific anxiety. While COVID-19 concern and non-specific anxiety are moderately correlated (Fig. 5c), we observe that they are associated with curiosity in substantially different ways. In contrast to the effect of COVID-19 concern, participants reporting high levels of non-specific anxiety were *less likely* to wait for answers of any sort (main effect $b = -0.28$, 95%

PI $= [-0.37, -0.20]$; Fig. 2a, Eqn. S6). This reduction in information-seeking parallels previous findings of the diminished pursuit of reward among people with depressed mood[44,56,57]. Furthermore, adding non-specific anxiety as a covariate did not alter the directing or energizing effects of COVID-19 concern (all results reported here are adjusted for non-specific anxiety).

The dissociation between COVID-19 concern and non-specific anxiety applied to usefulness judgments as well, with non-specific anxiety associated with a decrease in usefulness judgments $b = -0.10$, 95% PI $= [-0.15, -0.04]$, especially so for COVID-19-related

questions, as indicated by the interaction term for question type and non-specific anxiety $b = -0.07$, 95% PI $= [-0.12, -0.03]$ (Fig. 2b, Eqn. S10). Similarly, non-specific anxiety is associated with a reduction in satisfaction with answers of both types $b = -0.20$, 95% PI $= [-0.25, -0.15]$ (Fig. 3b, Eqn. S7).

Is the energizing effect of COVID-19 concern on information-seeking limited to questions conventionally considered useful, as intrinsic drive theories would predict? It is not: just like questions conventionally viewed as useful, seeking of answers to trivia questions is robustly predicted by COVID-19 concern; simple effect $b = 0.17$ 95% PI $= [0.07, 0.27]$, interaction $b = 0.01$, 95% PI $= [-0.03, 0.05]$. Seeking of answers to trivia questions is also predicted by judged usefulness (simple effect $b = 0.82$ 95% PI $= [0.57, 1.07]$), even more so than conventionally useful questions; interaction $b = 0.28$ 95% PI $= [0.11, 0.45]$. This demonstrates that utility maximization is an adequate explanation of curiosity for trivial knowledge, a result at odds with the current consensus that such curiosity is by-definition non-instrumental.

Finally, we tested whether our results generalize beyond the waiting task. See Supplementary Information and Fig. S6 for a successful conceptual replication using a self-report measure of curiosity.

**Motivation and subsequent learning**. A role for motivation and utility in driving information-seeking is rational only inasmuch as this relationship is manifested in subsequent learning. Indeed, we find that COVID-19 concern is associated with a directing effect on memory as well, as higher COVID-19 concern predicted poorer memory for general answers, but spared COVID-19-related answers (interaction $b = 0.06$, 95% PI $= [0.03, 0.09]$; Fig. 3c, Eqn. S8). Such a directing influence of motivation is akin to adaptive forgetting of motivationally irrelevant information[58,59]. This finding is also noteworthy as it suggests that COVID-19 concern is not merely a correlate of higher learning capacity, since higher COVID-19 concern is associated with remembering less general information.

Similar to information-seeking, the effects of COVID-19 concern on memory cannot be explained by mere anxiousness: we found no significant association between non-specific anxiety and memory $b = -0.04$, 95% PI $= [-0.08, 0.005]$ (Fig. 3d; Eqn. S8).

In general, we find that people tend to seek answers that they would subsequently be satisfied with and that they will remember (correlation between waiting and satisfaction $r = 0.60$, $n = 104$, two-sided $p = 1.26 \times 10^{-11}$; between waiting and memory $r = 0.39$, $n = 104$, two-sided $p = 4.85 \times 10^{-5}$; Fig. S3). Taken together, these findings indicate that motivation and utility explain not only information-seeking but are predictors of the full gamut of epistemic behavior, including subsequent processing of the information and long-term consequences for learning.

## Discussion

Together, these findings support the hypothesis that humans generally pursue useful information, even in situations heretofore considered driven by non-instrumental curiosity. Moreover, these results shed light on the psychological mechanisms of this process. We find that humans behave as if maximizing the expectation of informational utility, as derived by each individual from the content of a question according to their motivational state.

Deriving the exact personal utility of information based on one's motivation and goals is intractable computationally, and so must be approximated by any intelligent system[18,46]. We observe that the human cognitive system is largely rational in its approximation of personal utility. Thus, specific motivational states are associated with domain-specific differences in personal utility and information-seeking—the directing effect that is the hallmark of goal-rational behavior. However, these motivational states are also associated with the enhanced seeking of information of all content domains—a pattern predicted by an energizing role for motivation.

An energizing role is a basic tenet of theories of motivation[38,39], and has been repeatedly demonstrated in rats[38], yet our results join only a short list of findings in humans, and are the first outside the domain of simple reward-based behavior[60–62]. Furthermore, our data lend support to the hypothesis that motivation energizes information-seeking by altering the average expected utility of information. The use of average expected utility in decisions is a useful simplification from a computational perspective, but constitutes a deviation from purely normative behavior[38,39].

A change to average value wrought by COVID-19 concern likely happens over days and applies to a wide array of tasks. Presumably, motivation can energize information-seeking over a range of timescales and generalization, from the developmental scale, giving rise to stable personality traits, to the highly adaptive and local. Our experimental design was apt for causally demonstrating how trial-by-trial changes to the average value of information energize subsequent information-seeking. This finding complements previous work that ties these changes to value with subsequent memory[28]. Further elucidation of the types and limits of energizing effects awaits future research.

The notion that people use a grand average of the expected utility of information in their decisions can provide an explanation for a longstanding mystery regarding curiosity—that is people's tendency to seek more information than is economically prescribed[14,21,26]. This tendency is usually documented in paradigms specifically crafted to offer participants only what scientists consider to be useless information (a prime example being the outcome of random lotteries that would anyway be revealed later[21,26]). However, since the longer-term average value of information includes tasks where information is useful (that is, almost any other task in real life), participants making choices based on this long-term average would be biased to seek more information than is conventionally called for by the specifically crafted task.

It is common practice in curiosity research to assume that the pursuit of trivial knowledge cannot be driven by instrumental factors in any way, as the answers to these questions possess no utility[16,22,27,36]. Our data provide a clear empirical refutation of this assumption, as responses to trivia questions exhibited the same relationship with utility and motivation as did responses to questions considered instrumental. Our framework encompasses information-seeking across domains and predicts how and when motivated information-seeking in one domain may have a spill-over effect, driving information-seeking in another domain.

Our focus here was on the directing and energizing effects of COVID-19 concern. However, we also observed a main effect of non-specific anxiety on information-seeking, whereby more anxious people tended to seek less information. This dissociation between COVID-19 concern and non-specific anxiety was important to establish our main conclusions. Yet, this finding also has significance in its own right, as some previous work has found support for diminished curiosity in anxiety[57], while other researchers have postulated that anxiety might increase information-seeking, as it is associated with high intolerance for uncertainty[48,63,64]. We find a negative correlation between non-specific anxiety and information-seeking, a pattern mirroring established findings regarding diminished reward-seeking in anxiety and depression[44,56].

While the experiment reported here exhibits ecological validity —measuring information-seeking during the first wave of the pandemic—this validity comes at a cost of experimental control. Further research is needed to establish the directing and especially

the energizing role of motivation by causal manipulation of motivation in the lab. Direct manipulation is also essential for disentangling the effects of highly motivating and demotivating events, which were not separable in the unidimensional measure of COVID-19 concern. More than a century worth of research on the role of motivation in reward-based behavior should provide the appropriate tools to elicit shifts in motivational states and utilize the motivating power of stimuli.

The sudden onset of COVID-19 in spring 2020 afforded a unique opportunity to study information-seeking and motivation as they undergo rapid real-life changes. Yet, the uniqueness of circumstance might also hinder generalizing our inferences to behavior under more normal conditions. To mitigate this risk, we can use learning theory to analyze the onset of COVID-19 and produce a set of simple conditions an event must fulfill to elicit the patterns of behavior we observe. An event must cause a substantial and acute change to the life conditions of an individual, rendering a previously insignificant knowledge domain personally relevant. Such events commonly occur during an individual's lifetime: moving to a new town, enrolling in college, being drafted into the military, or becoming ill. Such events also occur on a societal scale, for example during important elections or natural disasters. It remains for future research to replicate and extend our findings focusing on such events.

More broadly, while we focused on how information-seeking is guided by informational utility, the amount of information expected from an answer should also be an important determinant of seeking it[33,45,46,65]. Estimating how much information would be gained from an answer is dependent on a person's existing knowledge about the question. This is especially pertinent in our case, since COVID-19 became motivationally relevant not only as a response to abrupt changes in the environment, but also through the knowledge regarding the virus we rapidly acquired during the early days of the pandemic.

The hypotheses presented here and the findings that support them strive to explain information-seeking using the same computational and algorithmic principles derived from studying reward-seeking behavior[22,28,35], rather than by postulating a special intrinsic drive. This framework is consistent with recent work implicating curiosity as a necessary core computation for natural and artificial intelligence[66,67], with information and reward postulated to be the basic fungible currencies of cognition[35,45,46]. Explaining ecological information-seeking in cost-benefit terms is a step towards understanding the economy of utility and information in the brain.

Finally, our findings address key open questions about how people seek information during the COVID-19 pandemic[41]. Whether people seek information rationally is the subject of active debate in public health and political science research[8,11,68,69]. Our findings suggest that humans are efficient information seekers, and that the measurement of motivational factors as modulators of expectations of utility is an important tool in understanding individual information-seeking behavior during a pandemic, or other events of interest for public policy. The energizing effect we document here could be of interest in situations where stimulating people's information-seeking is desirable—as is the case in public education campaigns, or in the classroom[70].

## Methods
**Data collection.** In all, 6135 participants were recruited through Amazon Mechanical Turk, with data collection occurring twice weekly, between 11 March

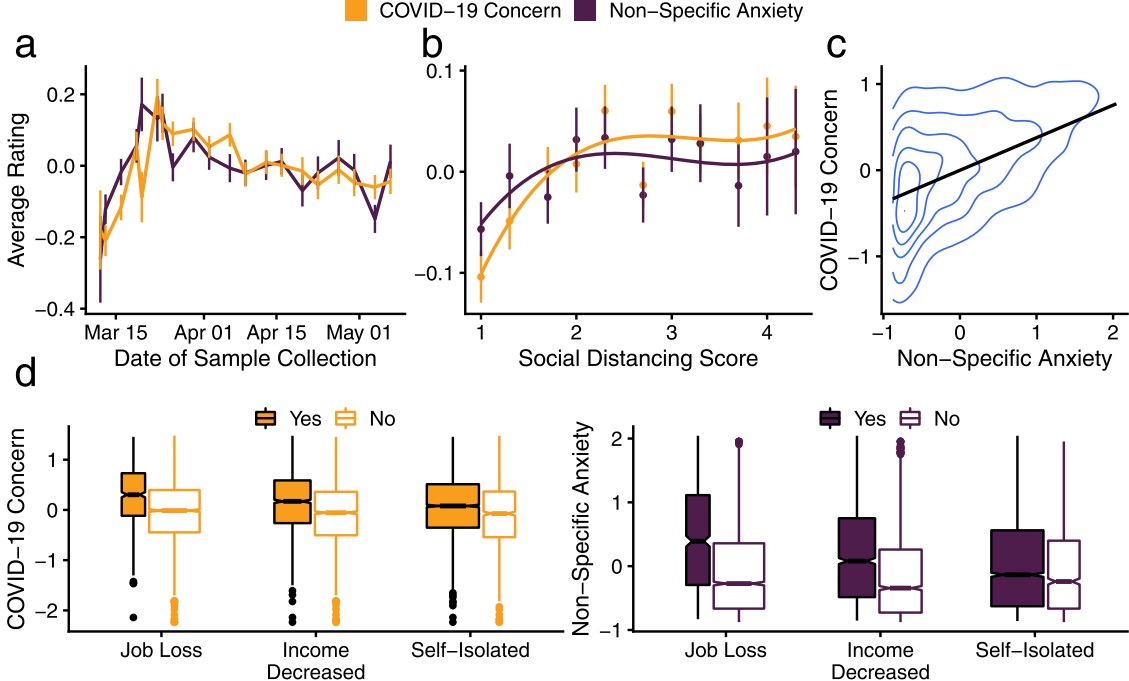

**Fig. 5 Motivational state metrics track real-world circumstances. a** Standardized rating of COVID-19 concern increased sharply mid-March 2020, a similar pattern is observed for non-specific anxiety. Data presented as mean values ±1 SEM; $n = 5375$ participants. **b** COVID-19 concern is higher for people in states where social distancing is practiced, as measured by Unacast via smartphone location data[75]. Data presented as mean values ±1 SEM, lines are predictions from a general additive model; $n = 5302$ participants. **c** Non-specific anxiety and COVID-19 concern are moderately correlated $r = 0.44$, $n = 5376$, two-sided $p = 2.16 \times 10^{-248}$; black line derived from linear regression. **d** COVID-19 concern levels (plotted on the left) are higher among participants who had lost their jobs, had seen their income decrease, or went into self-isolation. The same was true for specific anxiety, plotted on the right. Boxplots: central lines denote median, hinges mark the interquartile range (IQR), whiskers 1.5 times the IQR, width of box proportional to the square root of the number of participants; overall $n = 5375$ participants. Source data are provided as Source Data files.

and 7 May 2020. A week after the first session, participants were invited by email to complete a second session. Overall, 71.48% of eligible participants returned for session 2. Since we were interested in individual differences and given the uncertainty about the course of events in the early days of the pandemic, we opted for a maximal sample size given our budget: 400 participation slots were opened on each session in the first 5 weeks, and 300 thereafter. Data collection ceased with the full utilization of the budget.

All subjects provided informed consent; all protocols were approved by the Columbia University Institutional Review Board. Detailed data collection descriptions are available in Supplementary Information.

**Stimuli**. A set of questions and answers was used as stimuli in this experiment. 52 *COVID-19-related questions* were sourced from publications by the World Health Organization, US Centers for Disease Control and Prevention, or the New York Times. These were chosen such that half were clearly useful and half would be conventionally considered non-useful (usefulness was also assessed individually by the participants, see below). 52 *general questions* comprised the second type of questions—half of these were trivia questions drawn from previous studies[28], and half were useful household tips sourced from lists of tips on the internet. See supplementary material for a list of all questions.

A third category of crowdsourced general questions was included in the study for exploratory purposes and is not analyzed here—see Supplementary Information for details. Importantly, this category always appeared last.

**Task design**. As detailed below, we measured three aspects of epistemic behavior: *information-seeking*, *satisfaction*, and *memory*. We measured these constructs using tasks validated by previous studies of information-seeking, which did not address questions about motivation or utility[27,28,36]. As predictors of epistemic behavior under the framework discussed above, we measured *expectations of utility* and *motivational state*.

*Information-seeking* was quantified using the waiting task, based on participants' choices to wait for answers to presented questions. On each trial of the waiting task, participants were presented with a question, and three choice buttons. If they knew the answer to the question, they were instructed to press "know". Otherwise, they could choose to wait a specified duration for the answer by pressing "wait Xs" (4,8,12, or 16 s, randomly assigned), or else press the "skip" button, which terminated the trial. Importantly, participants knew that the entire duration of the task was independent of their choices (it was set to 300 s) and therefore were encouraged to use their own interest to decide whether to wait. The proportion of "wait" versus "skip" responses at variable waiting durations serves as our main index of information-seeking[28,36]. Following each presented answer, participants were asked to rate their *satisfaction* on a 1–5 Likert scale.

*Memory* for the answers was assessed 7–8 days later, using an answer recall task. On each trial of the recall task, participants were presented with a question they had chosen to wait for a week earlier. Participants indicated whether they remembered the answer to the question. If so, they had to input the answer into a text box. A research assistant blind to conditions and hypotheses coded the accuracy of these responses.

A measure of *utility expectations* was collected following the waiting task. Participants rated a randomly held out set of five questions of each type (COVID-19-related or general) on the expected usefulness of answers, both for themselves and for others, on a 1–7 Likert scale. Participants were not presented with the answers to these questions. We chose to measure the usefulness of questions that did not appear in the waiting task to prevent any bias resulting from responding twice to the same question.

*Motivational state* was assessed at the end of session 1. Participants completed a questionnaire probing their affective concerns regarding COVID-19. This questionnaire included items regarding anxiety about the medical, economic, and social circumstances, and perceptions of severity and risk (henceforth "COVID-19 concern"; Table S4 lists all questions). Prior to this, participants answered a series of questions regarding their non-specific anxiety and affective state (henceforth "non-specific anxiety"; Table S3), which allowed us to compare the effect of a domain-specific concern on information-seeking with the effect of anxiety in general. These two measures were validated by confirming their robust associations with real-life events such as self-isolation, job loss, or social distancing (see Fig. 5, Supplementary Information).

See Supplementary Information for complete timing parameters and Fig. S1 for screenshots of the tasks. The tasks were programmed using jsPsych 6.1.10.

**Analysis**. Our main goal of the analysis was to test the predictions described in the introduction regarding epistemic behavior. We first focused on the hypothesized effects of motivational state, namely COVID-19 concern, in interaction with attributes of the stimuli, on information-seeking. We then assessed the hypothesized link between COVID-19 concern and usefulness judgments. Lastly, we tested the whole framework, with a mediation analysis linking COVID-19 concerns to waiting for choices via usefulness judgments.

Data were analyzed using R 3.6.0, Stan 2.23.0. and Julia 1.4.2.

*Exclusion criteria*. Four participants reported technical difficulties in the presentation of questions. Their data were excluded from the analysis. Data from 358 participants (5.84%) reporting less than perfect English language fluency and 335 participants (5.46%) who interacted with other applications more than five times during the waiting or rating tasks were further excluded from the analysis. Following previous studies with the waiting task[28], we excluded data from participants who failed to respond on more than 20% of trials ($n = 4$, 0.07%), or whose mean response time was >2 standard deviations (SD) lower than the group average ($n = 58$, 0.95%). Overall, data from 5376 participants were included in analyses (median age 36, range 18–89; 2818 female).

We separately excluded data from the second session if participants had more than five application interactions during the recall task ($n = 176$, 3.27%), or gave responses that were not compliant with instructions (e.g., non-words, $n = 52$, 0.97%).

*Validation of motivational state measures*. Before addressing the role of motivational states in driving information-seeking, we first verified that our two affective measures (COVID-19 concern and non-specific anxiety) were internally consistent and related to relevant real-world experiences.

Ratings for the COVID-19 concern and non-specific anxiety questionnaires were subjected to a Bayesian Principal Component Analysis[71], which is robust to missing data. Fivefold cross-validation revealed that three was the optimal number of components in the data. The three components were rotated using the Quartimax method[72], and each item was assigned to the component on which it had the strongest loading. One group contained items related to COVID-19, comprising the COVID-19 concern measure, while another contained all items measuring negative affect, which we used as the non-specific anxiety control. The third component contained all positive affect items. We used the unweighted means of each variable group to avoid overfitting. The relation between affective measures and information-seeking are very similar when using a naive grouping of items, according to the original questionnaire they came from.

As expected, we found that both measures vary with events such as job loss, income reduction, self-isolation, social distancing behavior in the state, and the timeline of the virus spread in the United States (see Fig. 5, Supplementary Information). See the results above for the crucial divergence between COVID-19 concern and non-specific anxiety.

*Assessing the role of motivation*. Motivational Effects on Epistemic Behavior. To evaluate the two predicted effects of motivation on epistemic behavior, namely a directing effect and an energizing one, we constructed regression models predicting epistemic behavior at the single-trial level using COVID-19 concern and non-specific anxiety, in interaction with the type of question (COVID-19/general), and with judged usefulness for the question (averaged over the participants who rated each question). We also included wait duration as a covariate.

We used multilevel logistic regression to predict waiting and memory from the combination of these predictors, and multilevel ordered-logistic regression to predict satisfaction ratings (Eqn. S6–8).

Usefulness judgments, used as predictors in this set of regression models, were rated on an ordinal Likert scale. See Supplementary Information for details of the ordinal Item Response Model[73] we used in order to extract metric values for the usefulness predictor in the above regression models.

*Analyzing trivia questions*. To assess whether any conclusions we draw extend to trivia questions, an epistemic domain conventionally considered useless and hence beyond the remit of instrumental decision-making, we refit the above waiting choices model using only the general question block, with a question type predictor capturing differences between trivia questions and household tips.

*Motivational effects on judged usefulness*. We predicted that motivational effects on information-seeking operate by changing the expected utility of questions. Hence, we were interested to see whether mean usefulness judgment levels change with COVID-19 concerns. We fit a multilevel ordered-logistic regression model to usefulness judgments with the goal of estimating how usefulness judgments are influenced by COVID-19 concerns, with non-specific anxiety as a control covariate (Eqn. S10). We fit separate ordinal-regression threshold parameters for each usefulness item, to account for possibly different use of rating scales[73].

*Mediation model*. As a further test of our hypotheses, we determined whether the data are congruent with the notion that motivational states influence information-seeking by changing expectations of utility. While the data presented here cannot strictly support a causal account of such sort, joint statistical analysis of waiting choices and usefulness judgments can still validate whether the data conform with the predictions of the entire framework presented above. To validate these predictions, we fit a joint regression model (akin to mediation models), defined by two multilevel regression equations. The first equation defined the mediator model, predicting usefulness judgments from question type, COVID-19 concern, and the interaction of the two terms. The second equation defined the outcome model, predicting waiting from the same factors, with the addition of usefulness judgments and the covariate of wait duration. This joint model allowed us to estimate the extent to which the effects of COVID-19 concern and the interaction between

COVID-19 concern and question type on waiting are mediated by usefulness judgments. See Supplementary Information for a full description of the model (Eqn. S11–12), including how it was designed to account for the different sets of participants providing usefulness ratings and waiting for choices for each question.

*Prediction errors and subsequent waiting.* While our experimental design is largely correlational, it can provide causal evidence in support of participants' use of an average value of information in their waiting choices. Being influenced by an average value when making waiting choices entails being influenced by prior learning outcomes. To test this prediction, we estimated the effect of prediction errors elicited by the preceding trial on waiting choices in the subsequent trial. Following the reinforcement learning literature[51,52], we define the prediction error as the difference between actual satisfaction and usefulness expectations. We fit the data with a multilevel regression model predicting waiting on the subsequent trial from the satisfaction ratings and usefulness expectations on the preceding trial (Eqn. S13).

*Fitting regression models.* All multilevel regression models described above included maximal random effect structure and were fit to the data using Hamiltonian Monte-Carlo sampling implemented in the Stan language. Regularizing priors were used to facilitate estimation (Table S1). We report coefficients for all ratings-based predictors on a standardized scale. Four Monte-Carlo chains were run for each model, collecting 2000 samples each after a 1500 sample warmup period (for the mediation model 3000 samples were collected due to model complexity). Convergence was assessed using the $\hat{R}$ metric, and visual inspection of trace plots. For all models mentioned in the main text, R syntax formulae, as well as coefficients for covariates are reported in Supplementary Information.

**Reporting summary**. Further information on research design is available in the Nature Research Reporting Summary linked to this article.

## Data availability
The entire dataset discussed here has been deposited on the OSF database[74] https://osf.io/gjcu9/. Source data for figures are provided in this paper.

## Code availability
The code used to run the experiment as well as the scripts and software environment we used for analysis have been deposited on the OSF database[74] https://osf.io/gjcu9/.

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

## Acknowledgements
The authors thank Ohad Dan for their useful discussion and Nitai Kerem for his help with coding recalled answers. Funding was provided by a Templeton Foundation Science of Virtues grant (#60844 to D.S. and R.R.H.).

## Author contributions
Y.A., C.B.M., R.R.H., and D.S. designed research; Y.A. collected data; Y.A., C.v.G., and M.L. analyzed data; Y.A., R.R.H., and D.S. wrote the first draft; C.B.M., C.v.G., M.L., Y.A., R.R.H., and D.S. edited the paper; R.R.H. and D.S. supervised.

## Competing interests
The authors declare no competing interests.
