## [Peer Review File · Nature Communications]

An Energizing Role for Motivation in Information-Seeking During the Early Phase of the COVID-19 PandemicReviewers' comments:

Reviewer #1 (Remarks to the Author):

Abir and colleagues report a large online study taking advantage of the COVID-19 pandemic to investigate the interaction between concern regarding an unfolding global event, and human information seeking, valuation and memorisation. The manuscript is well written and relatively easy to follow.

The authors' working hypothesis is that concern about the topic (COVID-19) would lead to (1) increased topic-related information seeking due a specific increase in the utility of topic-related inquiries, and (2) a global increase in information-seeking due to a general "energising" effect on the utility of information seeking itself. To test the various components of this model, they used a simple delay-discounting task (measure of information seeking for COVID-19 vs. general information), combined with subjective ratings of the usefulness of questions and answers (measure of utility), questionnaire measures of anxiety and COVID-19 concern (measure of motivation and anxiety), and a memory recall test after one week.

The authors found that COVID-19 concern was associated with increased COVID-19-related information seeking relative to general questions ("directing effect"), but also increased seeking of general information ("energizing effect"). These two effects were mediated by the value attributed to COVID-19-related and general questions. Finally, COVID-19 concern was associated with a selective remembering of COVID-19-related information relative to general information. These findings are compatible with a view of ecological information-seeking as a utility maximization process. This type of framework is often adopted by experimental studies of information sampling (e.g., beads task, tile task, etc.) but its ecological validity is somewhat unclear. The present contribution is therefore interesting because of its ecological validity.

That said, my main concern is that I am not convinced that the rational framework proposed by the authors (directionality of the effects and components included) is the only one that can account for this set of results.

MAJOR ISSUES:

1. Causal claims unsupported by correlational data.

The data presented here fit well with the theoretical directional model put forward in Fig. 1a (i.e., "motivation" shapes utility, which in turn guides information seeking). My main concern, however, is that, because of the cross-sectional correlational nature of the data, none of the analyses performed in this work can provide support in favour of this specific model versus alternative ones in which the direction of some or all of the effects is reversed. For example, equally valid alternative interpretations of the authors' main findings could be that: (1) curious individuals who naturally seek more information were more likely to encounter COVID-19-related news during this period, and therefore became more concerned about this topic as a result (reversed accounts of the "energizing" effect); (2) individuals who particularly valued COVID-19-related information became more concerned about this topic as a result (reversed account of the "directing" effect); (3) individuals who more selectively encoded COVID-19 information into their long-term memory became more concerned about this topic as a result (reversed account of the "learning" effect). I can see that the authors were relatively careful not to make causal claims in the Results section, but anywhere else (abstract, introduction, discussion), the interpretation of the results is made in relation to a causal rational framework. In the absence of a solution to this chicken-and-egg problem, I am afraid that this contribution becomes much less interesting than it looks because the interpretation is highly speculative.

In the theoretical framework proposed in Fig. 1a, the very existence of a COVID-19-concern implies some form of feedback from the information gathered to the "motivation" component. How could "motivation" shape the question utility landscape if it is not informed by

From a methods perspective, the experimental design itself may have promoted some form of directional transfer from the waiting task to the measure of COVID-19 concern as the former systematically preceded the latter. It is therefore possible that the level of COVID-19 concern exhibited by participants may have been, at least in part, influenced by what they had learnt during the waiting task.

2. Potential role of COVID-19 uncertainty, knowledge and their variation over time.

At least two important potential confounds appear to have been neglected in the present work: COVID-19-related uncertainty and knowledge. Human information sampling behaviour is known to be sensitive to the current level of knowledge and uncertainty about the object of the search (e.g., <https://elifesciences.org/articles/12215>). As mentioned in my previous point, the measure of COVID-19 concern might, at least in part, reflect the degree of uncertainty and knowledge participants had about the pandemic. Because of the composite measure provided by the COVID-19 concern questionnaire, it is unclear the extent to which it is indeed the “motivation” and not the lack of knowledge and/or the level of uncertainty about the COVID-19 pandemic that is influencing information seeking behaviour in the current work.

Furthermore, as nicely illustrated in Fig. 4a, the average rating of COVID-19 concern was not stationary but instead increased sharply in March 2020 before to decrease slowly throughout April 2020. Thus, the participants who were the most concerned about COVID-19 correspond to those tested at the end of March, while the least concerned ones were tested earlier in March or at the end of April. During the same period of time, the authors report that the level of curiosity for COVID-19-related questions steadily decreased, likely due to an increase in knowledge about the pandemic in the general population (Supplementary Results). Is it therefore possible that the increase in knowledge and decrease in uncertainty account, at least in part, for the changes in COVID-19 concern and information seeking behaviour?

I am aware that the authors did not measure the level of knowledge/uncertainty about the pandemic but the waiting task could be used as an indirect proxy for COVID-19-expertise. If the average level of COVID-19-related knowledge increased in the general population, the proportion of COVID-19-related questions participants knew the answer to should increase over time.

From an analysis perspective, this indirect proxy of COVID-19-related expertise could be entered as an extra covariate in the regression model to test its relationship to information seeking behaviour. Another (imperfect) way to control for knowledge-related changes in COVID-19 concern would be to de-trend the measure of COVID-19-concern by z-scoring separately for each experimental sample rather than across the pooled dataset.

From a methods perspective, the measure of information seeking itself may be confounded by knowledge because of the design of the waiting task. This task consisted in a series of three-alternative forced choices (choice between “known answer”, “skip”, or “wait for the answer”) that had to be made within a predetermined amount of time (three minutes). The accuracy of the proportion of ‘wait’ vs. ‘skip’ responses (i.e., number of trials used to compute this proportion) therefore depended on the amount of knowledge an individual had. For example, a participant with a lot of COVID-19-related knowledge (potentially towards the end of the data collection period) may have wasted a lot of time responding that they knew the answer to the question, which left little time to measure the willingness to wait to learn unknown information. One way to minimise this potential confound would have been to discount the time wasted on questions participants knew the answer to from the three minutes of the task. I am unclear whether this was done or how this potential confound was minimised?

3. Inter-dependency across tasks.

If my understanding is correct, usefulness judgements and one-week recall were only performed on questions participants had chosen to wait for. Thus, once again, the accuracy of the measure of

usefulness (i.e. number of trials to average from) depended on participants' choices on the waiting task. Similarly, the proportion recalled was likely to be influenced by the quantity of information participants chose to be presented with during the waiting task. For example, a participant who chose to wait for 15 questions was less likely to remember the answer than one who only waited for five. How does the study/analysis control for these potential confounds?

4. Imprecise definition of motivation.

In the present work, COVID-19 concern and motivation are used interchangeably. However, there seems to be a missing conceptual step from one to the other. I found that it was quite hard to understand what the authors precisely meant by "motivation". The definition seemed to oscillate between 'preoccupations', 'worries', 'interest', 'behavioural relevance', 'goals', 'reasons to act', etc. Even the measure itself (questionnaire) is composite by nature, and seems to encompass various different concepts. The manuscript would benefit from a better definition of the concepts hypothesised to drive information seeking, rather than hypothesising that 'everybody knows what motivation is'.

MINOR ISSUES:

1. On several occasions, the text refers to the wrong figure/table. For example: p.8 line 161 4b should be 4c; p. 10 line 195 3c should be 3a; page 13 line 261 S2 should be S4.

2. Fig 1c does not capture the fact that the waiting task was a three- (and not two-) alternative forced choice. If my understanding is correct, participants first had to indicate whether they knew the answer to the question before to be offered the options to wait or skip. This is an important aspect of the task design that ought to be clearly reported.

3. The red and blue colours used in Figs 2 and 3 are indistinguishable when printed in black and white. The authors should use of redundant condition coding (e.g. red full line vs. blue broken line) to solve this issue.

4. The "Control Analyses" section of the Results (page 8) only reports the analysis of the proportion waited (Fig. 2a). For completeness, it should also report the analysis of usefulness judgement (Fig. 2b).

5. The control analyses (effect of non-specific anxiety) for the proportion recalled (Fig. 3b) and question satisfaction (Fig. 3d) are only reported in the legend of Fig. 3 but not in the main text. They should also be reported in the main text.

6. The authors report separate control analyses investigating the effect of non-specific anxiety but never control for non-specific anxiety in their main analyses. Do the main findings survive when controlling for non-specific anxiety in the regressions?

7. The main text reports that the duration of the waiting task was 180 secs (page 12, line 244) but the Supplementary Information reports 2.5 minutes (page 4, line 90).

8. When reporting the results in the main text, please refer to the corresponding regression equation in supplementary information.

9. When reporting statistical results in the main text, please include more information about what the parameter estimate represent (e.g. 'main effect of COVID-19 concern on waiting', 'Interaction COVID-19 x Question Type on waiting', etc.).

10. The Bayesian Principal Component Analysis of questionnaire data isolated three components (COVID-19 concern, negative affect, positive affect). The authors seem to have used the first two only. What was the relationship of positive affect to the other two questionnaire-derived measures, and to information seeking, valuation and memorisation?

11. In the discussion, the authors state “since the longer-term average of expected utility includes tasks where information is useful, participants making choices based on this long-term average would be biased to seek more information than is conventionally called for by current task” (page 18, lines 379-381). It would help the reader if you could provide an example of such scenario.

12. It is unclear whether the authors observed an interaction between COVID-19 concern and Question Type for the answer satisfaction (Fig. 3c).

13. I very much liked the visualisation of average COVID-19 concern and negative affect over time in Fig. 4. I suggest that the authors include the same type of visualisation – perhaps in Supplementary Information – for all variables considered (e.g., proportion waited for each question type, usefulness judgement for each question type, satisfaction for each question type, proportion recalled for each question type).

14. In the temporal sequence of the experiment (as well as in the rational framework), satisfaction comes before recall. I think that Fig. 3 c-d should come before Fig. 3 a-b.

15. In the introduction, the authors talk about the “failure of economic models of utility to predict human information seeking” (page 3, line 42-43). However, I would argue that there are many examples of successful use of neuroeconomics model to explain human information sampling behaviour (e.g., Furl et al., 2011 *J. Neurosci.*; Averbach 2015 *Plot Comput. Biol.*; Juni et al., 2015 *Decision*; Hauser et al., 2017 *Transl. Psychiatry*; etc.).

Reviewer #2 (Remarks to the Author):

The manuscript describes a single, large-scale study in which participants conducted at the beginning of the COVID pandemic, in which participants report on their wellbeing and their specific concern about COVID-19, and complete an information seeking task in which they are offered information related to the pandemic and non-relevant trivia. Responses are used to test novel claims about the rationality of human information seeking – that is, that motivation both directs and energizes information seeking. When an agent is motivated to learn about a particular topic, they will be more likely to seek information about that topic, but also about other, unrelated topics through a generalization process. This process is thought to happen through modulation of the expected utility of information – when one is motivated to learn, the expected utility of both relevant and non-relevant information increases.

I found the writing clear and compelling, and the topic is likely of interest to a broad audience. The results are potentially interesting, although I found the theoretical framing a bit of a stretch. I have a some comments and questions, mostly about the theoretical content and clarification points with regards to the method and results.

Introduction:

I am not convinced by the statement in the second paragraph of the introduction, characterising the “the prevalent conviction” is “that epistemic information-seeking is largely driven by non-instrumental curiosity”, and the development of the argument on page 5. In my understanding of the cited literature, it is recognised that much of human information seeking driven by utility, but that an additional process (curiosity) is needed to explain information seeking when there is little or no utility, or the information comes at a cost that outweighs its utility. This is why many authors (e.g. Lowenstein, 1994; Oudeyer & Kaplan, 2007) include internal (as opposed to external) motivation as a defining feature of curiosity (described as “the contemporary view” by Kidd and Hayden, 2016). Equally, experimentalists go to great lengths to design experiments in which the stimuli do not hold utility (hence the reliance on trivia questions, magic tricks, and gamble outcomes that are sure to be revealed later), to demonstrate that information seeking still occurs in the absence of utility. It seems somewhat unsurprising to me that people who are worried about COVID-19 will be more likely to seek

information about the topic than those who are ambivalent. I think I need more convincing that the literature suggests that people would not seek information in a way that that aligns with their goals, or a shift of focus to describing a mechanism by which the same process could explain both instrumental and non-instrumental information seeking (as the current study purports to do).

The authors use the term 'normative' in a number of places in the manuscript and I did not find it totally clear what they meant by this. For example, in the final paragraph on page 5, the authors refer to "the normative theories above", but I found it difficult to determine which of the above theories they are referring to. This makes that paragraph feel quite abstract and I struggled to keep track of the argument.

Learning/memory is not mentioned in the introduction until the description of the current study. Inclusion of literature on the effects of curiosity on learning/memory (e.g. Gruber et al., 2014), and on the generalisation of memory enhancements beyond the subject of the curiosity, would .

Method:

Is the measure of non-specific anxiety validated? I think it might be better described as a mood or wellbeing questionnaire since it includes feeling upset, angry, depressed, sad, and stressed, none of which are strictly characteristic of anxiety.

I am a little confused about the level of analysis for the main models – are the analyses conducted at the item/question level? If so, how does the incorporation of the expected utility work, since this is only collected for a few held out items. Were all the predictors at the participant level? I felt that this could have been explained much more clearly in the methods – in particular, on page 16, the authors state that "average judged usefulness for the question" was included in the models. Is this an error, or do the authors mean that the averaged usefulness from those participants who did judge this question was included?

Results:

In the analysis of recall of answers, was number of answers waited for controlled for? Could memory performance have been confounded by list length as COVID-19 concern increases? I don't think it would explain the pattern of results, but I do think it is worth controlling for.

Discussion:

The authors suggest that their findings demonstrate that human information seeking is indeed rational because it aligns with motivation. I am not sure that this interpretation follows from the results for two reasons. First, has the irrationality not just been moved to the motivation? How do we know whether an individual's motivation is rational? I think the authors can address this by referring to the 'validation of motivational state', but they may wish to do some more explicitly in terms of rationality. Second, since the energizing effect is apparently quite far-reaching, the individual who is rationally motivated to seek information about COVID-19, is also seemingly irrationally motivated to seek information about other topics, and equally, the individual who is not motivated to seek information about COVID-19 is not motivated to learn about other topics that might indeed be useful, like household tips.

It would be helpful to expand on the mechanism by which the expected utility is generalised. The authors describe an averaging process, but I struggle to understand what they mean by this. Do they mean across all possible information, all available information, the information they received in the recent past? Is the expected utility updated after receiving information as might be expected from a reward-learning model, or is there a separate process? Is the effect time-limited? Of course some of these may be empirical questions, but it would be informative for the authors to propose a mechanism that could be tested in future work.

The authors frame the importance of the study in terms of concern that non-rational information seeking leaves people vulnerable to nefarious sources of information. How do these findings address this problem, or do the authors believe that it isn't really a problem at all given their findings?

Reviewer #3 (Remarks to the Author):

Report on Behavior During the COVID-19 Pandemic Reveals the Rationality of Human Information-Seeking (NCOMMS)

This paper reports a huge MTurk survey as part of a broader study on curiosity, usefulness, etc. of information. Participants are shown questions about COVID and other, asked whether they will wait for answers (as a measure of expected information utility), and asked questions about usefulness, then satisfaction with answers, and later recall.

The paper is described as if it is a test of rational information-seeking. What is shown is that judged usefulness increases waiting (Fig S2) and people wait less with longer delays. These correlations indicate response to subjective benefit and cost. That is the minimal definition of rationality. The state of the art in computational modelling is to assess either monetary value (in a task where information value can be established) or some other numerical value scale, and the same with costs. Furthermore, another aspect of rationality is whether judgements of usefulness are accurate, and whether expected-to-be-useful information is then remembered better (otherwise it has limited benefit). There is no correlation reported between those measures. However, Figure 3 suggests satisfaction and recall are likely to be strongly negatively correlated. This is a bit puzzling— subjects are saying they are satisfied with a question answer, but then tend to forget it. Unless you do not think recall is part of the utility calculus for information (an important aspect of benefit), the full rationality interpretation is flimsy.

Other than the domain of COVID it was hard to figure out exactly what was brand new here. Have no previous experiments asked for ratings of “usefulness” (it is a central construct in (16) cited in main text)? One new aspect is the prediction about an energizing or spillover effect of curiosity in one domain (COVID) to another. Note that this does not appear to be causal, in the sense that changing COVID concern increases non-covid question concern. One wonders if you recruited these subjects, did not ask COVID questions, then just did the general questions whether you get the same results. If so it is not an energizing effect from curiosity about COVID (as created by the questioning).

I did not understand exactly what's in Figure 2c-d.

The abstract and introductory paragraphs contain a lot of very general, almost hyperbolic, claims. The title, for example, is over the top. You did not measure “behavior” (just MTurk answers) and as noted, you have established only a minimum (at best) concept of rational information-seeking.

Example (line 38): “This prevalent conviction, that human information-seeking is plagued by irrationality, is grounded in a long philosophical tradition and is reflected in a wide array of cognitive theories positing that epistemic information-seeking is largely driven by non-instrumental curiosity”. There is no convincing (peer-reviewed; cf citation (1) is not peer reviewed) evidence that this conviction is “prevalent”, that it is “plagued” (like a medicalization), what the philosophical tradition is, what the “wide array” is, or that non-instrumental is posited to be the “large[st]” driver of epistemic information-seeking. On the latter point, the authors might be mistaking the fact that people are interested in curiosity about non-instrumental information *because* it is not instrumental with the assertion that such motivations “largely drive” information seeking. (That is, an unusual behavior be overstudied relative to its everyday importance).

Line 53 “unique opportunity”. Every year there are interesting ongoing events which can be used (elections, natural disasters) similarly.

Line 58. Your study is not demonstrably ecologically valid, unless you take a further step to show that the questions you sampled are like those the general population actively seeks answers to.

The satisfaction question seems a little problematic. A person could be satisfied with an answer because they guessed correctly or because they were surprised (by their wrong guess) to learn

something new. And they could be unsatisfied if they thought the answer was too vague or wrong. You seem to need "satisfaction" to be an outcome where useful is a prediction.

There is no power calculation for this giant sample. Did you oversample, in the sense of gathering much more data than needed given planned hypotheses?

Reviewer 1

Comment 1

Reviewer 1 writes:

Abir and colleagues report a large online study taking advantage of the COVID-19 pandemic to investigate the interaction between concern regarding an unfolding global event, and human information seeking, valuation and memorisation. The manuscript is well written and relatively easy to follow.

The authors' working hypothesis is that concern about the topic (COVID-19) would lead to (1) increased topic-related information seeking due a specific increase in the utility of topic-related inquiries, and (2) a global increase in information-seeking due to a general "energising" effect on the utility of information seeking itself. To test the various components of this model, they used a simple delay-discounting task (measure of information seeking for COVID-19 vs. general information), combined with subjective ratings of the usefulness of questions and answers (measure of utility), questionnaire measures of anxiety and COVID-19 concern (measure of motivation and anxiety), and a memory recall test after one week.

The authors found that COVID-19 concern was associated with increased COVID-19-related information seeking relative to general questions ("directing effect"), but also increased seeking of general information ("energizing effect"). These two effects were mediated by the value attributed to COVID-19-related and general questions. Finally, COVID-19 concern was associated with a selective remembering of COVID-19-related information relative to general information. These findings are compatible with a view of ecological information-seeking as a utility maximization process. This type of framework is often adopted by experimental studies of information sampling (e.g., beads task, tile task, etc.) but its ecological validity is somewhat unclear. The present contribution is therefore interesting because of its ecological validity.

That said, my main concern is that I am not convinced that the rational framework proposed by the authors (directionality of the effects and components included) is the only one that can account for this set of results.

1. Causal claims unsupported by correlational data.

The data presented here fit well with the theoretical directional model put forward in Fig. 1a (i.e., "motivation" shapes utility, which in turn guides information seeking). My main concern, however, is that, because of the cross-sectional correlational nature of the data, none of the analyses performed in this work can provide support in favour of this specific model versus alternative ones in which the direction of some or all of the effects is reversed.

Thank you for this insightful comment. We agree that the cross-sectional nature of the experiment constrains the ability to draw causal inference from it. We addressed this point in

two ways. First, we expanded our discussion of this point (p. 19-20, 22-23) and amended any language that could be read as causal. Second, we supply new analyses and results. These address the question of causality by taking advantage of the temporal flow of the experiment and the fact that questions were presented in random order. This allowed us to ask whether there is any relationship between how participants feel about the answer to one question and their tendency to seek information on the next question. This analysis yielded causal evidence for the link between average utility (expressed as surprise, or prediction error) and information seeking. We now present this result on pp. 11-12, figure 4.

Additionally, since our findings encompass the gamut of epistemic behavior, rather than just information-seeking, we sought to address some of the alternative interpretations with the current data:

For example, equally valid alternative interpretations of the authors' main findings could be that: (1) curious individuals who naturally seek more information were more likely to encounter COVID-19-related news during this period, and therefore became more concerned about this topic as a result (reversed accounts of the "energizing" effect);

This is an interesting idea. While it is possible, however, it seems that this direction is rendered less plausible when considering the effect of COVID-19 concern not only on information-seeking, but also on memory. As now noted in the revised MS (p. 13), if the COVID-19 concern questionnaire merely picked up individuals who are more curious, a robust prediction would be that these individuals should also have better memory for the answers (see Gruber et al., 2014; Kang et al., 2009; Marvin & Shohamy, 2016). Instead, our data reveal the opposite pattern—overall, individuals who are high on COVID-19 concern were less likely to remember general knowledge (Fig. 3).

(2) individuals who particularly valued COVID-19-related information became more concerned about this topic as a result (reversed account of the "directing" effect);

Thank you. Yes, we agree on this point and, in fact, this is precisely the operational definition of motivation which we used (Dickinson & Balleine, 2002; Niv et al., 2006). We apologize for any confusion in our previous formulations, and in the revised MS we made sure to clarify the definition of motivational state and motivational shift in the introduction, as well as the exact nature of the predictions as they pertain to this definition (see p. 4 and 5 in the revised MS).

(3) individuals who more selectively encoded COVID-19 information into their long-term memory became more concerned about this topic as a result (reversed account of the "learning" effect).

This is another thoughtful suggestion, which the data can shed light on. In particular, as now noted in the MS (p. 13, figure 3), we think that the sign of the memory selectivity effect in our

data speaks against this alternative explanation. Selective memory for COVID-19 information manifested as increased forgetting of general knowledge answers. Since these answers were not known by participants prior to the experiment, selective forgetting of these answers could not have plausibly engendered COVID-19 concern.

I can see that the authors were relatively careful not to make causal claims in the Results section, but anywhere else (abstract, introduction, discussion), the interpretation of the results is made in relation to a causal rational framework.

Thank you for confronting us with this mistake. We closely reread these sections and edited out any causal language throughout the MS (e.g. p. 5, 8-13).

In the absence of a solution to this chicken-and-egg problem, I am afraid that this contribution becomes much less interesting than it looks because the interpretation is highly speculative.

We regret that we were not clear enough in our initial submission regarding this important point. As we clarify in the revised MS, our predictions are grounded in over 50 years of animal work regarding the roles of motivation in directing and energizing action. We elaborated the discussion of the long history of this theory, both in the introduction (p. 5) and in the discussion (p. 21), where we now specifically discuss the novel contribution of the results, the issue of causality as it bears out in our design and our results, and the tight link between theories in the field and how the new data we report shed new light on these theories.

In the theoretical framework proposed in Fig. 1a, the very existence of a COVID-19-concern implies some form of feedback from the information gathered to the “motivation” component. How could “motivation” shape the question utility landscape if it is not informed by

Thank you for raising this important point, we completely agree and now discuss the role of prior information in forming COVID-19 concern in the discussion section (page 23). In particular, we subscribe to the view that motivators acquire their potency via learning and that most motivated behavior is due to prior learning (Dickinson & Balleine, 2002). Epistemic behavior is no outlier – prior learning shapes utility, and so by definition motivates behavior. It is common in the motivation literature to examine one circuit of the learning-motivation-learning circle, and that is what we do in this manuscript. We agree with the point that theorizing about the full feedback circle makes for a fuller and more interesting perspective, as now noted in the revised MS.

From a methods perspective, the experimental design itself may have promoted some form of directional transfer from the waiting task to the measure of COVID-19 concern as the former systematically preceded the latter. It is therefore possible that the level of COVID-19

concern exhibited by participants may have been, at least in part, influenced by what they had learnt during the waiting task.

This is an interesting possibility, yet it seems somewhat less compatible with the data for two reasons. First, such a directional transfer should presumably have also affected the non-specific anxiety questionnaire, which was completed by participants prior to the COVID-19 concern questionnaire. Yet, the patterns linking COVID-19 concern and waiting are strikingly different from those linking non-specific anxiety and waiting. We now discuss this point in the Supplementary Information (p. 6).

Second, we replicate our main findings in the supplementary experiment, in which no answers were given to presented questions (participants rated their curiosity regarding answers, rather than choose to wait for answers). In this experiment learning would have been minimal, since no answers were given, and so presumably directional transfer is even less likely. We now note this fact in the Supplementary Information (p. 24).

Comment 2

At least two important potential confounds appear to have been neglected in the present work: COVID-19-related uncertainty and knowledge. Human information sampling behaviour is known to be sensitive to the current level of knowledge and uncertainty about the object of the search (e.g., <https://elifesciences.org/articles/12215>). As mentioned in my previous point, the measure of COVID-19 concern might, at least in part, reflect the degree of uncertainty and knowledge participants had about the pandemic. Because of the composite measure provided by the COVID-19 concern questionnaire, it is unclear the extent to which it is indeed the “motivation” and not the lack of knowledge and/or the level of uncertainty about the COVID-19 pandemic that is influencing information seeking behaviour in the current work.

This is an excellent point and it is indeed the case that in this work we focused on one of the two determinants of curiosity according to normative theory – utility, and its modulation by motivation. We view expectations of information gain as another important determinant, a potential factor of interest, rather than a confound. Forming expectations of information gain is closely related to current knowledge and uncertainty, and the literature linking these factors to information-seeking is rather developed, as aptly mentioned by the reviewer. We now elaborate on this issue in the discussion section, and postulate about possible interactions (page 23). We feel that addressing both factors is beyond the scope of a single manuscript, but knowledge is definitely part of our long-term research program. In addition, we followed your recommendations below for analyses on this current data that could speak to this issue.

Furthermore, as nicely illustrated in Fig. 4a, the average rating of COVID-19 concern was not stationary but instead increased sharply in March 2020 before to decrease slowly throughout April 2020. Thus, the participants who were the most concerned about COVID-19 correspond to those tested at the end of March, while the least concerned ones were tested earlier in March or at the end of April. During the same period of time, the authors report that the level

of curiosity for COVID-19-related questions steadily decreased, likely due to an increase in knowledge about the pandemic in the general population (Supplementary Results). Is it therefore possible that the increase in knowledge and decrease in uncertainty account, at least in part, for the changes in COVID-19 concern and information seeking behaviour?

I am aware that the authors did not measure the level of knowledge/uncertainty about the pandemic but the waiting task could be used as an indirect proxy for COVID-19-expertise. If the average level of COVID-19-related knowledge increased in the general population, the proportion of COVID-19-related questions participants knew the answer to should increase over time.

From an analysis perspective, this indirect proxy of COVID-19-related expertise could be entered as an extra covariate in the regression model to test its relationship to information seeking behaviour. Another (imperfect) way to control for knowledge-related changes in COVID-19 concern would be to de-trend the measure of COVID-19-concern by z-scoring separately for each experimental sample rather than across the pooled dataset.

Thank you for this suggestion, which prompted us to elaborate our models controlling for the linear and non-linear effects of time. We find that our conclusions generalize across all time points (see Supplementary Information, *Assessing the effect of time*). Additionally, we now include proportion 'Known' responses as an additional covariate in our models. Our main conclusions again hold on average even when adjusting for proportion 'Known'. Moreover, we see that participants who responded to more questions as *known* tended to wait less for the questions they did not know the answer to. This speaks against expertise with COVID-19 being a simpler explanation of our data (See Supplementary Information, *Assessing the effect of proportion 'Known' responses*, Fig. S4).

From a methods perspective, the measure of information seeking itself may be confounded by knowledge because of the design of the waiting task. This task consisted in a series of three-alternative forced choices (choice between "known answer", "skip", or "wait for the answer") that had to be made within a predetermined amount of time (three minutes). The accuracy of the proportion of 'wait' vs. 'skip' responses (i.e., number of trials used to compute this proportion) therefore depended on the amount of knowledge an individual had. For example, a participant with a lot of COVID-19-related knowledge (potentially towards the end of the data collection period) may have wasted a lot of time responding that they knew the answer to the question, which left little time to measure the willingness to wait to learn unknown information. One way to minimise this potential confound would have been to discount the time wasted on questions participants knew the answer to from the three minutes of the task. I am unclear whether this was done or how this potential confound was minimised?

Thank you for giving us the chance to clarify this important point. As now stated explicitly (Supplementary Information pp. 15-16) the proportion of questions participants responded to as ‘Known’ does change the number of waiting datapoints we have for each participant. It is worth to note that it does not bias waiting choices as an index of information-seeking, but rather only impacts the variance of our estimates. We do take this difference in variance into account – all our models allow for differing precision for each participant and each effect. As we use multilevel Bayesian regression modelling, this uncertainty is propagated to our posterior estimates. Hence, our inferences are robust to this feature of the task.

Comment 3

If my understanding is correct, usefulness judgements and one-week recall were only performed on questions participants had chosen to wait for. Thus, once again, the accuracy of the measure of usefulness (i.e. number of trials to average from) depended on participants’ choices on the waiting task.

We apologize for not being clear enough about this point. No, in fact participants rated the usefulness of a separate held-out set of questions. This set was determined randomly for each participant. No participant rated the usefulness of a question they had seen in the waiting task. We designed the experiment this way to avoid biasing either measure. We tried to explain this feature better in the Methods section (p. 15), and Supplementary Information (p. 4).

Similarly, the proportion recalled was likely to be influenced by the quantity of information participants chose to be presented with during the waiting task. For example, a participant who chose to wait for 15 questions was less likely to remember the answer than one who only waited for five. How does the study/analysis control for these potential confounds?

In principle, the number of answers participants chose to wait for could have influenced their proportion of recalled answers. Indeed, it is a very reasonable prediction. Fortunately, it is one that we can test in our data and which has been examined in previous studies as well. One of the astoundingly robust results of the curiosity literature is that when people select which answers to view, the number of answers they viewed, even though it sets the length of list to remember, does not predict memory performance (Gruber et al., 2014; Marvin & Shohamy, 2016; Tedeschi, 2020). To test this in our own results, we added the number of answers seen as a covariate to our memory models. We did not find that it affected the proportion of answers recalled, nor did it change any of the results in any way (see Supplementary Information, *Assessing the effect of motivational states on epistemic behavior*).

Comment 4

In the present work, COVID-19 concern and motivation are used interchangeably. However, there seems to be a missing conceptual step from one to the other. I found that it was quite hard to understand what the authors precisely meant by “motivation”. The definition seemed to oscillate between ‘preoccupations’, ‘worries’, ‘interest’, ‘behavioural relevance’, ‘goals’,

'reasons to act', etc. Even the measure itself (questionnaire) is composite by nature, and seems to encompass various different concepts. The manuscript would benefit from a better definition of the concepts hypothesised to drive information seeking, rather than hypothesising that 'everybody knows what motivation is'.

Thank you for stating this clearly. We agree and are grateful for the opportunity to have clarified our definition in the introduction (pp. 4,5). In the revised text, we define a motivational state as the mapping between potential actions and their value, and in the context of our study, the mapping between seeking-information in a specific domain, and the value of that information. We adopt this definition from learning theories of motivation, chief among them those formalized by Dickinson and Balleine (2002) and Niv et al. (2006).

We agree that the questionnaire measuring COVID-19 concern, which is the name we give to participants' COVID-19-specific motivational state, is a composite. We find that that is often the case with questionnaires. This compositional nature allows them to capture the many aspects of report about daily life. Since the questionnaire is a composite, we sought to validate it. We probed its internal consistency and relation to the other questionnaires using BPCA and reliability analyses, as well as relating COVID-19 concern to life occurrences (see the Methods section, pp. 17-18, and Supplementary Information, pp. 6-8, 17-18).

Minor comments

On several occasions, the text refers to the wrong figure/table. For example: p.8 line 161 4b should be 4c; p. 10 line 195 3c should be 3a; page 13 line 261 S2 should be S4.

Thank you for the attention you afforded our manuscript. We made sure to fix these typos and double-check all references to figures and other material.

Fig 1c does not capture the fact that the waiting task was a three- (and not two-) alternative forced choice. If my understanding is correct, participants first had to indicate whether they knew the answer to the question before to be offered the options to wait or skip. This is an important aspect of the task design that ought to be clearly reported.

Your understanding is very accurate. We amended the figure to better reflect the task, following your suggestion.

The red and blue colours used in Figs 2 and 3 are indistinguishable when printed in black and white. The authors should use of redundant condition coding (e.g. red full line vs. blue broken line) to solve this issue.

Thank you for bringing this issue to our attention. We changed the figures following your suggestion.

The “Control Analyses” section of the Results (page 8) only reports the analysis of the proportion waited (Fig. 2a). For completeness, it should also report the analysis of usefulness judgement (Fig. 2b).

We agree with your judgment, and now report the analysis of usefulness in this section (p. 12).

The control analyses (effect of non-specific anxiety) for the proportion recalled (Fig. 3b) and question satisfaction (Fig. 3d) are only reported in the legend of Fig. 3 but not in the main text. They should also be reported in the main text.

Thank you for noting this lack of consistency. These are now reported in the main text (pp. 12, 13).

The authors report separate control analyses investigating the effect of non-specific anxiety but never control for non-specific anxiety in their main analyses. Do the main findings survive when controlling for non-specific anxiety in the regressions?

Yes, they do! In fact, all the statistics we report regarding the main findings are from models adjusting for non-specific anxiety. We now clearly state so in the Results section (p. 12). These inferences also hold without adjusting for non-specific anxiety – see Supplementary Information.

The main text reports that the duration of the waiting task was 180 secs (page 12, line 244) but the Supplementary Information reports 2.5 minutes (page 4, line 90).

Thank you for finding this typo. The latter is correct, and the main text now lists the correct duration.

When reporting the results in the main text, please refer to the corresponding regression equation in supplementary information.

When reporting statistical results in the main text, please include more information about what the parameter estimate represent (e.g. ‘main effect of COVID-19 concern on waiting’, ‘Interaction COVID-19 x Question Type on waiting’, etc.).

These are good ideas. We implemented both throughout the text.

The Bayesian Principal Component Analysis of questionnaire data isolated three components (COVID-19 concern, negative affect, positive affect). The authors seem to have used the first two only. What was the relationship of positive affect to the other two questionnaire-derived measures, and to information seeking, valuation and memorisation?

We added a section to the supplement detailing these relationships (pp. 15, 17-18, 22). In short, positive affect is very negligibly correlated with COVID-19 concern and strongly anticorrelated with non-specific anxiety. When adding positive affect to our main model, all our previous inferences hold, and positive affect does not have any independent contribution to predicting information-seeking.

In the discussion, the authors state “since the longer-term average of expected utility includes tasks where information is useful, participants making choices based on this long-term average would be biased to seek more information than is conventionally called for by current task” (page 18, lines 379-381). It would help the reader if you could provide an example of such scenario.

Thank you for sharing the need for more clarity on this issue. We now elaborate on this issue in the discussion section (pp. 21-22). The prime example for such a scenario is tasks that involve participants paying a cost for revealing the outcomes of lotteries that would later be revealed anyway. By convention, this information has zero utility, as in the context of a lottery task learning the outcome of one lottery will not change your earnings on the next lottery. These tasks are thus specifically constructed to offer only information that is conventionally viewed as worthless. However, participants’ expectations of utility, according to the framework we present in the manuscript, are also affected by the longer-term average expectations of utility. These must be larger than zero, since entirely worthless information is very rare in real life. And so, participants choose to seek information in these tasks, even though the economic norm prescribes they should not.

It is unclear whether the authors observed an interaction between COVID-19 concern and Question Type for the answer satisfaction (Fig. 3c).

Yes, we do observe an interaction. People high in COVID-19 concern tend to be more satisfied with COVID-19-related answers relative to answers to general questions. This is discussed in the results section, under *Answer Satisfaction Reflects Motivational Changes to Utility* (p. 10).

I very much liked the visualisation of average COVID-19 concern and negative affect over time in Fig. 4. I suggest that the authors include the same type of visualisation – perhaps in Supplementary Information – for all variables considered (e.g., proportion waited for each question type, usefulness judgement for each question type, satisfaction for each question type, proportion recalled for each question type).

Thank you for the positive comment and for this suggestion! We added this as figure S5.

In the temporal sequence of the experiment (as well as in the rational framework), satisfaction comes before recall. I think that Fig. 3 c-d should come before Fig. 3 a-b.

Thank you for bringing this inconsistency to our attention. We amended the figure as you recommended.

In the introduction, the authors talk about the “failure of economic models of utility to predict human information seeking” (page 3, line 42-43). However, I would argue that there are many examples of successful use of neuroeconomics model to explain human information sampling behaviour (e.g., Furl et al., 2011 J. Neurosci.; Averbeck 2015 Plot Comput. Biol.; Juni et al., 2015 Decision; Hauser et al., 2017 Transl. Psychiatry; etc.).

We completely agree. We should have been more precise and strived for greater accuracy about this point in the revised introduction (p. 3). While economic models are successful up to a point, a consensus has formed that economic models of utility are not sufficient in explaining unconstrained information-seeking (note that all these interesting references regard tasks that set a clear goal for information-seeking). Indeed, some operational definitions of curiosity see it as the superfluous tendency to seek information beyond what is prescribed by these economical models.

Reviewer 2

Introduction-related comments:

The manuscript describes a single, large-scale study in which participants conducted at the beginning of the COVID pandemic, in which participants report on their wellbeing and their specific concern about COVID-19, and complete an information seeking task in which they are offered information related to the pandemic and non-relevant trivia. Responses are used to test novel claims about the rationality of human information seeking – that is, that motivation both directs and energizes information seeking. When an agent is motivated to learn about a particular topic, they will be more likely to seek information about that topic, but also about other, unrelated topics though a generalization process. This process is thought to happen through modulation of the expected utility of information – when one is motivated to learn, the expected utility of both relevant and non-relevant information increases.

I found the writing clear and compelling, and the topic is likely of interest to a broad audience. The results are potentially interesting, although I found the theoretical framing a bit of a stretch. I have a some comments and questions, mostly about the theoretical content and clarification points with regards to the method and results.

I am not convinced by the statement in the second paragraph of the introduction, characterising the “the prevalent conviction” is “that epistemic information-seeking is largely driven by non-instrumental curiosity”, and the development of the argument on page 5. In my understanding of the cited literature, it is recognised that much of human information seeking driven by utility, but that an additional process (curiosity) is needed to explain

information seeking when there is little or no utility, or the information comes at a cost that outweighs its utility. This is why many authors (e.g. Lowenstein, 1994; Oudeyer & Kaplan, 2007) include internal (as opposed to external) motivation as a defining feature of curiosity (described as “the contemporary view” by Kidd and Hayden, 2016). Equally, experimentalists go to great lengths to design experiments in which the stimuli do not hold utility (hence the reliance on trivia questions, magic tricks, and gamble outcomes that are sure to be revealed later), to demonstrate that information seeking still occurs in the absence of utility. It seems somewhat unsurprising to me that people who are worried about COVID-19 will be more likely to seek information about the topic than those who are ambivalent. I think I need more convincing that the literature suggests that people would not seek information in a way that that aligns with their goals, or a shift of focus to describing a mechanism by which the same process could explain both instrumental and non-instrumental information seeking (as the current study purports to do).

Thank you for raising this important point and for driving us towards greater precision. This comment dovetails with reviewer 1’s last comment. As you well noted, while utility models are successful up to a point, a consensus has formed in the literature that models of utility are not sufficient in explaining unconstrained information-seeking. Indeed, some operational definitions of curiosity see it as the superfluous tendency to seek information beyond what is prescribed by these economical models.

Hence, we reoriented our introduction towards your suggestion (p. 3), highlighting that our data suggest that there is no need for a separate mechanism to explain what has often been known as non-instrumental information seeking. We also expanded the literature we refer to when establishing our main claims. References 1-12 demonstrate said conviction in the wider, public context, while references 13-26 illustrate both examples of constrained information-seeking being driven by utility, and failures of utility to explain unconstrained information-seeking, or theories advancing the need to construe curiosity as a goal-independent drive.

The authors use the term ‘normative’ in a number of places in the manuscript and I did not find it totally clear what they meant by this. For example, in the final paragraph on page 5, the authors refer to “the normative theories above”, but I found it difficult to determine which of the above theories they are referring to. This makes that paragraph feel quite abstract and I struggled to keep track of the argument.

Thank you for noting this. We agree that this term can be ambiguous. We edited the text throughout to be more precise and specific, rather than use umbrella terms such as normative.

Learning/memory is not mentioned in the introduction until the description of the current study. Inclusion of literature on the effects of curiosity on learning/memory (e.g. Gruber et al., 2014), and on the generalisation of memory enhancements beyond the subject of the curiosity, would .

We quite agree and are grateful for your pointing this out. In fact, curiosity being so beneficial for memory is one of the reasons to suspect that it is always instrumental. We now discuss that early in the introduction (p. 3).

Methods comments:

Is the measure of non-specific anxiety validated? I think it might be better described as a mood or wellbeing questionnaire since it includes feeling upset, angry, depressed, sad, and stressed, none of which are strictly characteristic of anxiety.

Thank you for raising this issue. Indeed, the questions in the non-specific anxiety questionnaire come from well-validated and established questionnaires (STAI: Spielberger, 1983; and the Gallup wellbeing questionnaire). We agree that it could also be described as mood. Our purpose in labeling it non-specific anxiety was to highlight its role in our analysis as a control for the mood/anxiety components of COVID-19 concern. These composite questionnaires, while displaying internal consistency and reliability, are composites, and so were not easy to label. We now add a comment addressing this issue (see Supplementary Information, p.7).

I am a little confused about the level of analysis for the main models – are the analyses conducted at the item/question level? If so, how does the incorporation of the expected utility work, since this is only collected for a few held out items. Were all the predictors at the participant level? I felt that this could have been explained much more clearly in the methods – in particular, on page 16, the authors state that “average judged usefulness for the question” was included in the models. Is this an error, or do the authors mean that the averaged usefulness from those participants who did judge this question was included?

Thank you for pushing us to be clearer about these important details. We tried to add clarity in the revised methods section and Supplementary Information. As now explained in detail (p. 18-19), analysis is conducted at the level of the single trial. Since participants did not rate the usefulness of questions they were presented with in the waiting task, we used estimates from all the other participants who did rate those questions as predictors. The resulting usefulness predictor may be less powerful, as it ignores the idiosyncrasies of the participant making the waiting choice. However, this separation of rating from choosing allowed us to be confident that our measurements were not biasing each other.

Results comments:

In the analysis of recall of answers, was number of answers waited for controlled for? Could memory performance have been confounded by list length as COVID-19 concern increases? I don't think it would explain the pattern of results, but I do think it is worth controlling for.

We now include this analysis (see Supplementary Information, pp. 10, 19). In principle, memory performance could have been confounded by list length -- a very reasonable prediction.

Fortunately, it is one that we can test in our data and which has been examined in previous studies as well. One of the astoundingly robust results of the curiosity literature is that when people select which answers to view, the number of answers they viewed, even though it sets the length of list to remember, does not predict memory performance (Gruber et al., 2014; Marvin & Shohamy, 2016; Tedeschi, 2020). To test this in our own results, we added the number of answers seen as a covariate to our memory models. We did not find that it affected the proportion of answers recalled, nor did it change any of the results in any way.

Discussion comments:

The authors suggest that their findings demonstrate that human information seeking is indeed rational because it aligns with motivation. I am not sure that this interpretation follows from the results for two reasons. First, has the irrationality not just been moved to the motivation? How do we know whether an individual's motivation is rational? I think the authors can address this by referring to the 'validation of motivational state', but they may wish to do some more explicitly in terms of rationality. Second, since the energizing effect is apparently quite far-reaching, the individual who is rationally motivated to seek information about COVID-19, is also seemingly irrationally motivated to seek information about other topics, and equally, the individual who is not motivated to seek information about COVID-19 is not motivated to learn about other topics that might indeed be useful, like household tips.

Thank you for this insightful comment. Indeed, we should have been more careful in applying the term *rational* and explaining exactly what we mean. It is no mere triviality that participants are rational given their motivation, since it has been an open question whether humans can carry out the computations necessary to follow their goals when seeking information without constraints. Indeed, that is the premise behind positing a utility-divorced drive such as curiosity. We now expand on this topic in the introduction, explain the non-normativity of the energizing drive in the discussion, and refrain from claiming rationality without explanation (e.g. pp. 3, 21). We also reformulated the title to be more specific in this regard.

It would be helpful to expand on the mechanism by which the expected utility is generalised. The authors describe an averaging process, but I struggle to understand what they mean by this. Do they mean across all possible information, all available information, the information they received in the recent past? Is the expected utility updated after receiving information as might be expected from a reward-learning model, or is there a separate process? Is the effect time-limited? Of course some of these may be empirical questions, but it would be informative for the authors to propose a mechanism that could be tested in future work.

Thanks for pushing us on this point – it inspired us to be more precise and it led us to consider additional predictions regarding our data. Building on Niv et al. (2006), we do expect the average value to be learnt via a reward-learning model. We tested this prediction with a new analysis of prediction errors (see p. 11 and figure 4), the results of which provide evidence

supporting this account. We now elaborate on the proposed mechanism in the discussion section as well (p. 21).

The authors frame the importance of the study in terms of concern that non-rational information seeking leaves people vulnerable to nefarious sources of information. How do these findings address this problem, or do the authors believe that it isn't really a problem at all given their findings?

This is an important and interesting point, thank you. Most concretely, we think that while this study sheds light on this idea with empirical data, our findings don't suggest a direct solution to this problem yet. But, as we now stated in the revised discussion (p. 23), they do suggest a principled framework for running interventional studies in the future. We suggest that measuring motivation in tandem with information-seeking is crucial for understanding this problem. Additionally, consulting a century of research on motivation in the domain of rewards suggests many candidates for intervention.

Reviewer 3

This paper reports a huge MTurk survey as part of a broader study on curiosity, usefulness, etc. of information. Participants are shown questions about COVID and other, asked whether they will wait for answers (as a measure of expected information utility), and asked questions about usefulness, then satisfaction with answers, and later recall.

The paper is described as if it is a test of rational information-seeking. What is shown is that judged usefulness increases waiting (Fig S2) and people wait less with longer delays. These correlations indicate response to subjective benefit and cost. That is the minimal definition of rationality. The state of the art in computational modelling is to assess either monetary value (in a task where information value can be established) or some other numerical value scale, and the same with costs.

Thank you for this thoughtful comment, which dovetails nicely with some of the questions raised by R1 and R2. We appreciate the opportunity to clarify our thinking on this central issue. Indeed, we should have been more careful in applying the term *rational* and explaining exactly what we mean.

It is true that, as you noted, people show cost-benefit rationality in our data, modulating their waiting according to usefulness and wait duration. A more central claim in our paper is that they are also goal-rational, that is rational given their motivation. It is no mere triviality that participants are rational given their motivation, since it has been an open question whether humans can carry out the computations necessary to follow their goals when seeking information without constraints. Indeed, the lack of such ability is the premise behind positing a utility-divorced drive such as curiosity. We now expand on this topic in the introduction (p. 3). We also reformulated the title to be more specific in this regard.

Note that our models that quantify these effects can be used to express the effect sizes we find in terms of wait-duration-equivalent, in units of seconds. This would tether every finding to the numerical value scale of opportunity cost. At the time, we found this expression of effect sizes less intuitive for the reader than the standardized scale. If the editor and reviewers see this as a worthy addition, we are happy to do so.

Furthermore, another aspect of rationality is whether judgements of usefulness are accurate, and whether expected-to-be-useful information is then remembered better (otherwise it has limited benefit).

Thank you. We agree that there are potentially many different aspects of rationality that our data can speak to. Although accuracy of the usefulness expectations is difficult to judge in principle, our data do show that these ratings are remarkably consistent, evidenced by the fact that usefulness ratings measured in a separate group of participants predicts waiting behavior in a robust manner (this is due to the experimental design separating usefulness ratings from waiting choices).

Moreover, as you predicted, our data do indeed demonstrate that information that is expected to be useful is remembered better, as shown in Fig. S2c.

However, figure 3 suggests satisfaction and recall are likely to be strongly negatively correlated. This is a bit puzzling— subjects are saying they are satisfied with a question answer, but then tend to forget it. Unless you do not think recall is part of the utility calculus for information (an important aspect of benefit), the full rationality interpretation is flimsy

We apologize for not being clear enough about the relationship between satisfaction and memory. In fact, satisfaction and recall are positively correlated, as shown in Fig. S3 and as discussed in the Results section (under *Motivational Effects on Subsequent Learning*). As we write in the introduction (p. 3), results (p. 13), and discussion (p. 23), we absolutely share your view that memory for information is a central part of the utility calculus.

Other than the domain of COVID it was hard to figure out exactly what was brand new here. Have no previous experiments asked for ratings of “usefulness” (it is a central construct in (16) cited in main text)?

Thank you for pointing out the difficulty in gleaning the novelty from our writing. We tried to state it more clearly. In brief, and as now stated explicitly (pp. 5, 21, and noted also in the letter to the editor above), there are several novel aspects to the paper: this is the first study measuring information-seeking in tandem with expectations of usefulness and motivational states and the first demonstration of an energizing role for motivation outside the domain of reward-based behavior. We are especially proud of the ecological validity of our findings, having measured information-seeking in the early days of the pandemic. The onset of the pandemic set-up a natural experiment in this regard, engendering profound motivational shifts

that exposed the roles motivation plays in epistemic behavior. Indeed, the energizing role we found for motivation, while postulated over half a century ago and well established in animal work, has only been sporadically demonstrated in humans.

Regarding your comment – yes, indeed, our measurement usefulness in the context of curiosity and information-seeking is novel – and has only been done in one paper previously, regarding a specific kind of information-seeking which is quite different from what we explore here (Liquin and Lombrozo, 2020). Dubey and Griffith’s paper, which we rely on a great deal, is mostly concerned with the amount of information an answer supplies, rather than the kind of information and its relation to a person’s motivation. Moreover, the theoretical discussion of utility in that paper does not supply any relevant data nor an empirical demonstration of the link between utility and information-seeking. Indeed, a central drive for the current study was to test whether the theoretically hypothesized link between information-seeking and utility bears out in measurable, real-world behavior.

One new aspect is the prediction about an energizing or spillover effect of curiosity in one domain (COVID) to another. Note that this does not appear to be causal, in the sense that changing COVID concern increases non-covid question concern. One wonders if you recruited these subjects, did not ask COVID questions, then just did the general questions whether you get the same results. If so it is not an energizing effect from curiosity about COVID (as created by the questioning).

Thank you for noting that the energizing effect is novel and exciting. Yes, our data is mostly correlational, however, it is worth emphasizing that the block order was counterbalanced, therefore, we assume that the spill-over, or energizing effect as we term it, is not attributable to the order of questions in our experiment. Rather, we believe that it reflects the relationship between motivational state and information-seeking. We discuss this issue now, as well as the broader question of causality, in greater length in the revised discussion section (p. 21), and above in this letter.

Moreover, we now have causal support for the energizing effect. Using new analyses and results, we address the question of causality by taking advantage of the temporal flow of the experiment and the fact that questions were presented in random order. This allowed us to ask whether there is any relationship between how participants feel about the answer to one question and their tendency to seek information on the next question. This analysis yielded causal evidence for the link between average utility (expressed as surprise, or prediction error) and information seeking. We now present this result on pp. 11-12, figure 4.

I did not understand exactly what’s in Figure 2c-d.

We are sorry we were not clear enough and are grateful for this opportunity to clarify, as this is a central figure for the paper. These panels show the results of a joint analysis of motivation, usefulness, and waiting, showing evidence in support of the central hypothesized relationship between them. We have now expanded and clarified our description of the panels in the

revised MS and also describe them in length in the methods section (p. 19) and the supplement (pp. 10-11).

The abstract and introductory paragraphs contain a lot of very general, almost hyperbolic, claims.

We hope you'll find the revised version much more conservative and qualified and we thank you for the comment.

The title, for example, is over the top. You did not measure "behavior" (just MTurk answers) and as noted, you have established only a minimum (at best) concept of rational information-seeking.

We are thankful for the candid and direct feedback, however, with all due respect, we are also a bit baffled by it. We do not read the title as claiming anything about the entire domain of behavior. We merely qualify the claim in the title, by saying that it is supported by behavioral data (and, as such, is novel relative to prior theoretical claims about utility and information-seeking). We do believe that responses to our task, whether collected on MTurk or otherwise, qualify as behavior. We are not quite sure what other term would more accurately capture the data succinctly. Regarding the concept of rationality – please see replies above. It is quite challenging to write a title that captures the essence of the findings and framework in just a few words. This is something we put a great deal of thought in to and we would be open to consider alternative suggestions.

*Example (line 38): "This prevalent conviction, that human information-seeking is plagued by irrationality, is grounded in a long philosophical tradition and is reflected in a wide array of cognitive theories positing that epistemic information-seeking is largely driven by non-instrumental curiosity". There is no convincing (peer-reviewed; cf citation (1) is not peer reviewed) evidence that this conviction is "prevalent", that it is "plagued" (like a medicalization), what the philosophical tradition is, what the "wide array" is, or that non-instrumental is posited to be the "large[st]" driver of epistemic information-seeking. On the latter point, the authors might be mistaking the fact that people are interested in curiosity about non-instrumental information *because* it is not instrumental with the assertion that such motivations "largely drive" information seeking. (That is, an unusual behavior be overstudied relative to its everyday importance).*

We now discuss all these more thoroughly, with much expanded references. We think the characterization of the information-seeking literature, especially the subfield interested in unconstrained information-seeking, holds. In the revised MS, you'll find references 1-12 demonstrating said conviction in the wider, public context. References 13-26 illustrate both examples of constrained information-seeking being driven by utility, and failures of utility to

explain unconstrained information-seeking, or theories advancing the need to construe curiosity as a goal-independent drive

Line 53 “unique opportunity”. Every year there are interesting ongoing events which can be used (elections, natural disasters) similarly.

A year and a half into the pandemic, we must say that this comment took us by surprise. While we agree there are interesting ongoing events, many are unpredictable, and most – if not all - are not as all-encompassing as a pandemic, nor do most events offer such a unique angle on the acquisition of novel and unfolding information as it relates to most people’s day-to-day experiences across the country and the world. We believe our effort is unique, at least in the sense that no one has done it before. We hope that it will not be unique in the future, meaning, that follow-up studies will ensue along the lines of your prescient comment.

Line 58. Your study is not demonstrably ecologically valid, unless you take a further step to show that the questions you sampled are like those the general population actively seeks answers to.

Ecological validity is, of course, a relative matter. We are proud that our study is considerably more ecologically valid than average cognitive science study. We were happy to learn that reviewers 1 and 2 agreed with this judgment. We sampled our questions from the New York Times, and the website of the CDC. These were indeed high-traffic websites during the period of data collection, perhaps evidence in support of the general population seeking such information.

The satisfaction question seems a little problematic. A person could be satisfied with an answer because they guessed correctly or because they were surprised (by their wrong guess) to learn something new. And they could be unsatisfied if they thought the answer was too vague or wrong. You seem to need “satisfaction” to be an outcome where useful is a prediction.

Thank you for this comment and for the opportunity to clarify this measure, its background, and our interpretation of it. First, we wish to clarify that this measure draws on previously published work where it was established as a reliable and relevant measure of interest for assessing both information-seeking and memory (Marvin and Shohamy, 2016). As noted there and in the revised MS, our measurement of satisfaction is an outcome in the sense that it is measured after reading an answer to a question and it quantifies a subjective experience. In that sense, it is akin to other behavioral outcomes, such as memory. Importantly, here usefulness was measured in relation to the question alone, without reading the answer, hence it is a prediction. You raise many important factors that could add to variability in satisfaction ratings – as is true for many other psychological variables that assess a subjective experience. Yet, in addition to the previously published findings regarding the same measurement of satisfaction,

we also find robust effects of usefulness predictions and motivational states on satisfaction, even on top of all these potential sources of noise.

There is no power calculation for this giant sample. Did you oversample, in the sense of gathering much more data than needed given planned hypotheses?

Yes. Given the tight timetable of the project, we did not have time for a simulation to determine sample size for multilevel regression models fit to the waiting task, usefulness ratings, and motivational questionnaires. Note that measuring all those in tandem is a key novelty of our study, and therefore no prior work existed upon which to base power calculations. We specified big sample sizes as we describe in the methods section, choosing to err on the side of too much power rather than too little. Notably, we also conceptually replicated these findings in a much smaller independent sample, as described in the Supplementary Information (pp. 24-26).

Reviewers' comments:

Reviewer #1 (Remarks to the Author):

Abir and colleagues provided an extensive revision that has improved the manuscript overall. The revised introduction is more precise - in particular when it comes to the definition of the concepts used – and the interpretation of the results more careful. The authors have successfully addressed most of my points but I still have some concern about my first comment. The numbers below refer to the comment numbers of my previous review.

As a side note for future work, I would advise the authors to paste revisions in their response letter and highlight changes in the revised manuscript to facilitate the review process.

Comment 1

Thank you for using a more careful language and for explicitly discussing what can and cannot be received as causal evidence.

I can see that the authors have added an interesting new analysis (effect of question usefulness and answer satisfaction of trial n on the probability of waiting in trial $n+1$) in response to my point about causality. The rationale is that, if information seeking is influenced by the average utility of information seeking and the latter is updated on a trial-by-trial basis, then prediction errors (PE) in information utility should influence information seeking from one trial to the next. More specifically, positive PE should increase information seeking while negative PE should decrease it. This is a valid hypothesis that fits well within the theoretical framework used by the authors. However, I am not sure that it can readily be tested here given the experimental paradigm.

If my understanding is correct, although answer satisfaction was collected on a trial-by-trial basis in the waiting task, question usefulness was only collected afterwards, on a separate held-out set of questions (5 questions per category, i.e. 10 in total; p. 15). I am therefore a bit unclear as to how the authors used the measure of question usefulness in a trialwise analysis... My understanding is that they actually used the average usefulness rating of a given question type (COVID-related or general, i.e. only 2 values per participant) for all trials of the same type.

The problem with this approach is that the variable “question usefulness” can be used as a mere proxy of question type (given that most individuals rated COVID-19 related questions as more useful than general ones, and Eq S13 included a random effect of participant). Thus, another way to interpret the result of the x-axis of Fig. 4 is that participants were less likely to wait for the answer when the previous trial was COVID-related, potentially because they were more likely to have waited already (since they waited more for COVID-related question than general questions according to Fig. S5a for example). In other words, this finding might be confounded by the fact that participants were unwilling to wait twice in a row.

Next, the authors derive PEs as the difference between satisfaction and usefulness, standardised within each question type (Fig. 4a). For the same reason stated above, this metric is likely to in fact reflect almost exclusively answer satisfaction (as opposed to being equally influenced by both question usefulness and answer satisfaction). In summary, although the analysis of previous answer satisfaction seems valid, the other two (question usefulness and PE) are problematic.

The author's main answer to the alternative interpretations I suggested is that the pattern of memory performance seems at odd with such reversed accounts. This is a relatively valid point. However, using a similar reasoning, one could argue that the pattern of memory performance is in fact also at odd with the framework defended by the authors. In Figs. 2a and 3a it looks as though what drives the key interaction is the unwillingness to wait for COVID-19 questions and the dissatisfaction with COVID-19 responses in individuals who are the least concerned about COVID-19 (left side of the graph). Nevertheless, the effect on memory appears to be driven by a propensity to forget general

information in individuals who are the most concerned about COVID-19 (right side of the graph). According to the authors' framework, my understanding is that differences in recall (between general and COVID-19 related information) should be largest where differences in utility are maximal, i.e. in individuals with low COVID-19 concern. It is unclear to me how the authors explain this discrepancy.

Comment 2

Thank you for providing a more transparent account of the effect of time and previous knowledge on your task.

Comment 3

Supplementary Information mentions a control analysis in which the number of questions waited was added as a covariate (Eq S9, p. 10) but I cannot find the results of this analysis in the manuscript or Supplementary Information. Please add.

Comment 4

Thank you for explicitly stating in the manuscript the definition you used for 'motivation'. This has improved the clarity of the text.

Reviewer #2 (Remarks to the Author):

I feel that most of my comments have been dealt with satisfactorily in the revised manuscript. This is an interesting dataset and some novel ideas are explored in the analysis.

With regards to the additional work exploring the energising effect, these findings align closely with the reward-learning account of knowledge acquisition proposed by Murayama and colleagues (2019) in which they similarly predict the generalisation of expected value of information via prediction errors (<https://doi.org/10.1007/s10648-019-09499-9>).

I must say that I am still not entirely convinced by the labelling of the "non-specific anxiety" measure and I do not really understand why the authors would not just use a measure of anxiety if this was their aim.

Finally, I spotted that there is a formatting error in the table on page 50 of the supplementary materials so that the response scale and correlation with scale mean are missing for one item.

Reviewer #3 (Remarks to the Author):

This is a fairly responsive revision. However, I am just not a fan of this paper (though the other referees are). In general I think there has been far too much insta-science trying to learn generalizable lessons from the pandemic which is such an unusual event. Because it is so unusual, the usual inferences about causality, as well as generalizability, are not well-founded. To be clear, this is a criticism not only of your research but also of hundreds of other social science papers which are trying to learn from this unprecedented (and highly confounded) event. In any case, my pessimistic view should not keep this paper from moving forward at Nature Comms, and it is good to have some interesting data on this topic.

We address the reviewers' comments and suggestions below. Changes to the manuscript are marked in blue font.

Reviewer 1

Comment 1

Reviewer 1 writes:

Thank you for using a more careful language and for explicitly discussing what can and cannot be received as causal evidence.

I can see that the authors have added an interesting new analysis (effect of question usefulness and answer satisfaction of trial n on the probability of waiting in trial $n+1$) in response to my point about causality. The rationale is that, if information seeking is influenced by the average utility of information seeking and the latter is updated on a trial-by-trial basis, then prediction errors (PE) in information utility should influence information seeking from one trial to the next. More specifically, positive PE should increase information seeking while negative PE should decrease it. This is a valid hypothesis that fits well within the theoretical framework used by the authors. However, I am not sure that it can readily be tested here given the experimental paradigm.

If my understanding is correct, although answer satisfaction was collected on a trial-by-trial basis in the waiting task, question usefulness was only collected afterwards, on a separate held-out set of questions (5 questions per category, i.e. 10 in total; p. 15). I am therefore a bit unclear as to how the authors used the measure of question usefulness in a trialwise analysis... My understanding is that they actually used the average usefulness rating of a given question type (COVID-related or general, i.e. only 2 values per participant) for all trials of the same type.

The problem with this approach is that the variable "question usefulness" can be used as a mere proxy of question type (given that most individuals rated COVID-19 related questions as more useful than general ones, and Eq S13 included a random effect of participant). Thus, another way to interpret the result of the x-axis of Fig. 4 is that participants were less likely to wait for the answer when the previous trial was COVID-related, potentially because they were more likely to have waited already (since they waited more for COVID-related question than general questions according to Fig. S5a for

example). In other words, this finding might be confounded by the fact that participants were unwilling to wait twice in a row.

Next, the authors derive PEs as the difference between satisfaction and usefulness, standardised within each question type (Fig. 4a). For the same reason stated above, this metric is likely to in fact reflect almost exclusively answer satisfaction (as opposed to being equally influenced by both question usefulness and answer satisfaction). In summary, although the analysis of previous answer satisfaction seems valid, the other two (question usefulness and PE) are problematic.

Thank you for this thoughtful comment. Below and in the manuscript (Supplementary Information, p. 13) we now clarify our explanation of the prediction error model and elaborate on why the analysis allows for separation of the usefulness and satisfaction ratings from question type.

To start with, yes, it is correct that one set of participants rated a given question on its usefulness, and a different set of participants made waiting choices for that question. This was an important feature of the design to avoid the possible confound of participants explicitly trying to be consistent with their usefulness ratings when making waiting choices. Accordingly, we averaged the usefulness ratings for each question across all the participants who rated it and used this average as a predictor in the PE model. Hence, importantly, the usefulness predictor in the model has a different value for each question. It does neglect the idiosyncrasies of the participant making waiting choices, instead assuming that the group average can approximate their unobserved ratings. It is, if you wish, an inherent out-of-sample test of the model. This is the same situation for the simpler models we report earlier, and they serve as a validation of this approach.

The important point is that both usefulness and satisfaction vary per question, and so are not conflated with question type. Furthermore, COVID-19-related questions and general questions were administered in separate blocks in the waiting task, so the trial-by-trial dynamics we report cannot be ascribed to the effect of question type, especially since block order is counterbalanced.

We added an explanation to this effect to the section in the Supplementary Information explaining the model. Following the model equation, we now state: *Importantly, both the previous usefulness ratings and the previous satisfaction ratings in this equation vary on a trial-by-trial basis, and so are adequate for assessing the trial-by-trial influence of PEs on waiting choices. While satisfaction was rated by the same participant making the waiting choices constituting the outcome measure for this model, usefulness ratings in this equation, as in previous models, are the group average for the set of participants that rated each question. We are assuming that ratings elicited from other participants can predict waiting behavior, an assumption largely vindicated by our previous analyses using question usefulness. Moreover, each type of question - COVID-19 related questions and general questions - was administered in separate blocks of the waiting task, thus ensuring that any trial-by-trial effects are not confounded by question type.*

The author's main answer to the alternative interpretations I suggested is that the pattern of memory performance seems at odd with such reversed accounts. This is a relatively valid point. However, using a similar reasoning, one could argue that the pattern of memory performance is in fact also at odd with the framework defended by the authors. In Figs. 2a and 3a it looks as though what drives the key interaction is the unwillingness to wait for COVID-19 questions and the dissatisfaction with COVID-19 responses in individuals who are the least concerned about COVID-19 (left side of the graph). Nevertheless, the effect on memory appears to be driven by a propensity to forget general information in individuals who are the most concerned about COVID-19 (right side of the graph). According to the authors' framework, my understanding is that differences in recall (between general and COVID-19 related information) should be largest where differences in utility are maximal, i.e. in individuals with low COVID-19 concern. It is unclear to me how the authors explain this discrepancy.

Thank you for pushing our reasoning on this and giving us a chance to clarify our interpretation of this result – we find the discussion of our findings insightful and stimulating. We amended our claim in the revised manuscript (p. 13) and clarify below exactly what kind of inference we have license to make.

It seems to us that the plots and models in fact do not reliably indicate whether the COVID-19 concern effect is driven by high- vs. low-COVID-19-concern individuals. For all the predicted variables in figures 2 and 3 (waiting choices, usefulness ratings, satisfaction ratings and memory) we used a (generalized) linear model linking COVID-19 concern to outcome. Hence, our models give the same weight to participants across the COVID-19 concern spectrum, as COVID-19 concern has but one linear coefficient in the model. We'd like to suggest thinking about these plots in terms of differences in intercept *and* differences in slope between COVID-19-related and general questions. When both exist, there is an appearance of inhomogeneity in driving the effect, but in fact the model is agnostic to that.

The model does tell us that COVID-19-related question and general questions are waited for in similar proportions, on average (Fig. 2a). This gives the *appearance* of a rather symmetric effect for the COVID-19 concern slope. Furthermore, COVID-19-related questions are perceived as more useful on average (Fig. 2b). This gives the impression that the COVID-19 concern effect on usefulness is driven by high-COVID-19 concern individuals. For satisfaction (Fig. 3a) – the answers to COVID-19-related questions were less satisfying than general ones (perhaps because they were less definite), and so the COVID-19 concern effect appears to be driven by low-COVID-19-concern individuals. However, it is important to emphasize that these appearances cannot be substantiated by the models or the plots.

Following your comment, we realized that our writing imprecisely referred to differences between low- and high-COVID-19-concern individuals. Accordingly, we amended the relevant text in the results section to precisely reflect the inference that can be drawn from our model. Instead of referring to low- vs. high-COVID-19 concern individuals, we now write: *This finding is also noteworthy as it suggests that COVID-19 concern is not merely a correlate of higher learning capacity, since higher COVID-19 concern is associated with remembering less general information.*

It is an interesting question whether the effect of COVID-19 concern is homogenous, or whether there are actually distinct groups along the COVID-19 concern spectrum. To test for such an inhomogeneity of the COVID-19-concern effect we would have to develop some class of non-linear models. We worry that given our initial hypotheses, non-linear models would be much more ad-hoc and run the risk of over-fitting, relative to linear models.

Your thoughtful comments highlight the limitations of our dataset, which we now expanded in the discussion section (following your thoughtful comments on the first round, as well; pp. 21-22). Ultimately the answer to these questions will come from manipulating motivation under experimental control. We now highlight this in the discussion section, writing that *Direct manipulation is also essential for disentangling the effects of highly motivating and demotivating events, which were not separable in the unidimensional measure of COVID-19 concern.*

Preliminary causal support to our framework can be found in our prediction error data, while the prediction error effects on memory published previously (e.g. Marvin and Shohamy, 2016) also provide complimentary evidence regarding this issue. We now discuss the latter in the appropriate section of the discussion (see additions on page 21).

Comment 2

Thank you for providing a more transparent account of the effect of time and previous knowledge on your task.

Thank you for suggesting it!

Comment 3

Supplementary Information mentions a control analysis in which the number of questions waited was added as a covariate (Eq S9, p. 10) but I cannot find the results of this analysis in the manuscript or Supplementary Information. Please add.

The results of this analysis are reported in Supplementary Information p. 19: *“When adding the number of answers waited for as a covariate to this model, results are virtually unchanged. The number of answers waited for is not a significant predictor of memory, $b=0.04$, 95% PI = [-0.004, 0.08], with the sign of the coefficient opposite of that predicted by a list-length effect. Such an independence of memory from list-length has already been observed in studies of curiosity (Gruber et al., 2014; Marvin & Shohamy, 2016). The effects of question usefulness, COVID-19 concern and non-specific anxiety on memory, as well as the interactions of these factors with question type, remain unchanged when adjusting for number of answers waited for.”*

Comment 4

Thank you for explicitly stating in the manuscript the definition you used for ‘motivation’. This has improved the clarity of the text.

Thank you for giving us feedback on this! We tried to be as clear as we could, and we’re happy to hear that the result is satisfactory.

Reviewer 2

Reviewer 2 writes:

I feel that most of my comments have been dealt with satisfactorily in the revised manuscript. This is an interesting dataset and some novel ideas are explored in the analysis.

With regards to the additional work exploring the energising effect, these findings align closely with the reward-learning account of knowledge acquisition proposed by Murayama and colleagues (2019) in which they similarly predict the generalisation of expected value of information via prediction errors (<https://doi.org/10.1007/s10648-019-09499-9>).

Thank you for introducing us to this insightful paper. We agree – our results align well with its predictions, and with the general approach to studying information-seeking. We added a reference to this paper in the section introducing the prediction error analysis (reference number 54 on p. 11).

I must say that I am still not entirely convinced by the labelling of the "non-specific anxiety" measure and I do not really understand why the authors would not just use a measure of anxiety if this was their aim.

Thank you for this important comment. We admit that labeling this dimension was not easy, a point we mention in the Supplementary Information (p. 7). In the end, after extensive reading and discussion, we feel that "non-specific anxiety" is the best label we could find, given that the questions comprising it come from the STAI – an established anxiety questionnaire. It also includes questions that are very similar from the Gallup Wellbeing questionnaire. The main use of this scale is to control for any chronic anxiety or negative affect that is *not due specifically* to COVID-19. We are not wed to this label and are entirely open to other suggestions. We merely sought a label that would both be true to the source of the measure, on one hand, while capturing its contrast from the COVID-19-specific concern measure which it is meant to control for, on the other hand.

Finally, I spotted that there is a formatting error in the table on page 50 of the supplementary materials so that the response scale and correlation with scale mean are missing for one item.

Thank you for affording our work such close attention! We fixed the table.

Reviewer 3

Reviewer 3 writes:

This is a fairly responsive revision. However, I am just not a fan of this paper (though the other referees are). In general I think there has been far too much insta-science trying to learn generalizable lessons from the pandemic which is such an unusual event. Because it is so unusual, the usual inferences about causality, as well as generalizability, are not

well-founded. To be clear, this is a criticism not only of your research but also of hundreds of other social science papers which are trying to learn from this unprecedented (and highly confounded) event. In any case, my pessimistic view should not keep this paper from moving forward at Nature Comms, and it is good to have some interesting data on this topic.

We appreciate and understand your frank critique of our work. We agree that the uniqueness of the situation might limit inference and generalizability to everyday life. To mitigate this risk, we sought to clearly state the conditions needed to replicate and extend our findings. In doing so, we note that unusual events are often unusual from the point of view of a single individual but are common life experiences shared by many people, even if only once in their lifetime. The onset of COVID-19 was unique in the sense that it motivated us to observe the behavioral patterns we report and made it possible for us to observe them at a large scale. We hope COVID-19 is the only pandemic of our lifetime, but also expect to experience more acute life-changes. We hope to leverage such future events to continue studying the role of motivation in information-seeking.

In thinking this through more and in response to your generous and thoughtful comment, we have added a point about this to the discussion section (p. 23):

The sudden onset of COVID-19 in spring 2020 afforded a unique opportunity to study information-seeking and motivation as they undergo rapid real-life changes. Yet, the uniqueness of circumstance might also hinder generalizing our inferences to behavior under more normal conditions. To mitigate this risk, we can use learning theory to analyze the onset of COVID-19 and produce a set of simple conditions an event must fulfil to elicit the patterns of behavior we observe. An event must cause substantial and acute change to the life conditions of an individual, rendering a previously insignificant knowledge domain personally relevant. Such events commonly occur during an individual's lifetime: moving to a new town, enrolling in college, being drafted into the military, or becoming ill. Such events also occur on a societal scale, for example during important elections or natural disasters. It remains for future research to replicate and extend our findings focusing on such events.

Reviewers' comments:

Reviewer #1 (Remarks to the Author):

Many thanks for addressing my last remaining concerns in your revision. I now better understand how the authors used usefulness ratings in a trial-by-trial analysis, and judge this approach satisfactory given the study design. Regarding whether specific sub-groups of individuals along the COVID-concern spectrum drive the various interactions, I am glad to see that the authors are cautious about what can and cannot be inferred on the basis of the linear models used. I am happy with the authors' response and amendments to the text, and would encourage them to follow-up on this point in future work.

I have no further comment at this point, other than Fig 2 seems to have reverted back to the original version in the revised manuscript (dotted blue lines are missing).

We once again wish to thank the reviewers for the considerable and insightful attention you have afforded our work. We are happy that you find it interesting and worthy of publication.

We are grateful to reviewer 1 for their keen attention – we made sure the revised version of figure 2 is included in the submission